# Abundance and Sources of Atmospheric Halocarbons in the Eastern Mediterranean

Fabian Schoenenberger[1], Stephan Henne[1], Matthias Hill[1], Martin K. Vollmer[1],
Giorgos Kouvarakis[2], Nikolaos Mihalopoulos[2], Simon O'Doherty[3], Michela Maione[4],
Lukas Emmenegger[1], Thomas Peter[5], Stefan Reimann[1]

[1]Laboratory for Air Pollution/Environmental Technologies, Empa, Swiss Federal Laboratories for Materials Science and Technology, Dübendorf, Switzerland
[2]Department of Chemistry, University of Crete, Heraklion Crete, Greece
[3]School of Chemistry, University of Bristol, Bristol, UK
[4]Department of Pure and Applied Sciences, University of Urbino, Urbino, Italy
[5]Institute for Atmospheric and Climate Science, ETH Zürich, Zürich, Switzerland

*Correspondence to*: Stephan Henne (stephan.henne@empa.ch)

## Abstract

A wide range of anthropogenic halocarbons is released to the atmosphere, contributing to stratospheric ozone depletion and global warming. Using measurements of atmospheric abundances for the estimation of halocarbon emissions on the global and regional scale has become an important top-down tool for emission validation in the recent past, but many populated and developing areas of the world are only poorly covered by the existing atmospheric halocarbon measurement network. Here we present six months of continuous halocarbon observations from Finokalia on the island of Crete in the Eastern Mediterranean. The gases measured are the hydrofluorocarbons (HFCs), HFC-134a ($CH_2FCF_3$), HFC-125 ($CHF_2CF_3$), HFC-152a ($CH_3CHF_2$) and HFC-143a ($CH_3CF_3$), and the hydrochlorofluorocarbons (HCFCs), HCFC-22 ($CHClF_2$) and HCFC-142b ($CH_3CClF_2$). The Eastern Mediterranean is home to 250 million inhabitants, consisting of a number of developed and developing countries, for which different emission regulations exist under the Kyoto and Montreal Protocols. Regional emissions of halocarbons were estimated with Lagrangian atmospheric transport simulations and a Bayesian inverse modelling system, using measurements at Finokalia in conjunction with those from Advanced Global Atmospheric Gases Experiment (AGAGE) sites at Mace Head (Ireland), Jungfraujoch (Switzerland) and Monte Cimone (Italy). Measured peak mole fractions at Finokalia showed generally smaller amplitudes for HFCs than at the European AGAGE sites, except periodic peaks of HFC-152a, indicating strong upwind sources. Higher peak mole fractions were observed for HCFCs, suggesting continued emissions from nearby developing regions such as Egypt and the Middle East. For 2013, the Eastern Mediterranean inverse emission estimates for the four analysed HFCs and the two HCFCs were 13.9 (11.3-19.3) Tg $CO_2$eq $yr^{-1}$ and 9.5 (6.8-15.1) Tg $CO_2$eq $yr^{-1}$, respectively. These emissions contributed 16.8% (13.6-23.3%) and 53.2% (38.1-84.2%) to the total inversion domain, which covers the Eastern Mediterranean as well as Central and Western Europe. Greek bottom-up HFC emissions reported to the UNFCCC were higher than our top-down estimates, whereas for Turkey our estimates agreed with UNFCCC-reported values for HFC-125 and HFC-143a, but were much and slightly smaller for HFC-134a and HFC-152a, respectively. Sensitivity estimates suggest an improvement of the a posteriori emission estimates, i.e. a reduction of the uncertainties by 40-80% in the entire inversion domain, compared to an inversion using only the existing Central European AGAGE observations.

# 1 Introduction

Anthropogenic halocarbons, i.e. chlorofluorocarbons (CFCs), hydrochlorofluorocarbons (HCFCs), hydrofluorocarbons (HFCs), halons and other brominated species, are used in a wide range of industrial and domestic applications (e.g., refrigeration, air conditioning, foam blowing, solvent usage, aerosol propellants and fire retardants). Whereas only chlorinated and brominated halocarbons are responsible for stratospheric ozone depletion, most long-lived halocarbons are potent greenhouse gases [*Carpenter and Reimann*, 2014; *Farman, et al.*, 1985; *Molina and Rowland*, 1974; *Myhre, et al.*, 2013].

Ozone depleting substances (ODSs) are regulated by the Montreal Protocol (MP), which resulted in the global phase-out of CFCs from emissive use by 2010. HCFCs, which serve as transitional replacement products, are subject to a less demanding multistep phase-out ending in 2030 for Non-Article-5 (developed) and 2040 for Article-5 (developing) countries [*Braathen, et al.*, 2012]. To track the development of CFCs and HCFCs, the MP requires signatory parties to produce an inventory of their ODS consumption and production [*McCulloch, et al.*, 2001].

HFCs, used as second-generation replacement products for ODSs, do not contain chlorine or bromine. However, as some of them have a large global warming potential (GWP) and a projected rapid increase in their emissions, HFCs may significantly contribute to global radiative forcing as a direct consequence of protecting the ozone layer [*Montzka, et al.*, 2015; *Rigby, et al.*, 2014; *Steinbacher, et al.*, 2008; *Velders, et al.*, 2012]. HFCs are addressed within the Kyoto Protocol to the United Nations Framework Convention on Climate Change (UNFCCC). Signatory parties with binding emission reduction targets (Annex I) are required to submit their HFC emission inventories to the UNFCCC [*UNFCCC*, 1997]. These inventories are based on statistical "bottom-up" estimates, using production and consumption data and have been suspected to carry significant uncertainties [e.g., *Keller, et al.*, 2012; *Levin, et al.*, 2010; *Lunt, et al.*, 2015; *Rigby, et al.*, 2014]. In 2016, HFCs were included in the Montreal Protocol by the Kigali Amendment targeting a step-wise phase down of global consumption.

To validate reported inventories, "top-down" approaches, based on atmospheric measurements and atmospheric transport and chemistry models can be used. The combination of observations with simplified global scale box-models, allows the independent derivation of global emissions [e.g., *Carpenter and Reimann*, 2014; *Rigby, et al.*, 2010; *Schoenenberger, et al.*, 2015; *Vollmer, et al.*, 2015]. The application of more detailed atmospheric models has proven to be a powerful tool to quantify emissions on a spatially and temporally more explicit level enabling for

emission estimates on a continental to country scale [*Brunner, et al.*, 2012; *Ganesan, et al.*, 2014; *Graziosi, et al.*, 2015; *Hu, et al.*, 2015; *Keller, et al.*, 2012; *Kim, et al.*, 2010; *Lunt, et al.*, 2015; *Maione, et al.*, 2014; e.g., *Manning, et al.*, 2003; *Saikawa, et al.*, 2012; *Stohl, et al.*, 2009].

In Europe, the AGAGE network provides high-frequency observations of atmospheric halocarbons at 3 sites: Mace Head (Ireland), Zeppelin mountain (Spitsbergen, Norway), Jungfraujoch (Switzerland) and the affiliated station at Monte Cimone (Italy) [*Prinn, et al.*, 2000]. While data from this network have been frequently used in top-down estimates of Western European halocarbon emissions [e.g., *Brunner, et al.*, 2012; *Keller, et al.*, 2012; *Reimann, et al.*, 2008], the network has a very limited sensitivity towards emission from Eastern European sources (Figure 1a). For Eastern European HFC emissions, the importance of extending the observational network was illustrated by the large discrepancies between bottom-up emissions reported to UN-FCCC and those estimated top-down in an inverse modelling study using atmospheric observations obtained during a field campaign at K-Puszta in Hungary [*Keller, et al.*, 2012].

Even less reliable information on halocarbon emissions is available from the Eastern Mediterranean region, comprising Turkey, which is regarded as a developing country in the terminology of the Montreal Protocol (Article 5) but is a signatory party with binding emission reduction targets under the Kyoto Protocol (Annex I), Non-Article 5/Annex I states such as Greece, Romania, Bulgaria and Cyprus, and developing economies (Article 5/Non Annex I) such as Egypt and Israel with less stringent regulations and reporting requirements.

Estimating halocarbon emissions by top-down methods in the Eastern Mediterranean gains additional importance in the light of the beginning phase-out of HCFC emissions in Article-5 countries under the Montreal Protocol. This motivated our halocarbon measurement campaign at Finokalia (Crete, Greece) from December 2012 to August 2013. Here, we present the observed atmospheric halocarbon levels and combine the dataset with halocarbon observations at Jungfraujoch, Mace Head and Monte Cimone, atmospheric transport modelling and a Bayesian inversion system to derive the first comprehensive top-down emission estimates of HFC-134a ($CH_2FCF_3$), HFC-125 ($CHF_2CF_3$), HFC-152a ($CH_3CHF_2$), HFC-143a ($CH_3CF_3$), HCFC-22 ($CHClF_2$) and HCFC-142b ($CH_3CClF_2$) in the Eastern Mediterranean.

## 2   Methods

### 2.1      Observational Sites

Halocarbon measurements were conducted from December 2012 to August 2013 at the atmospheric observation site in Finokalia (FKL, 35.34°N, 25.67°E, 250 m a.s.l. [*Mihalopoulos, et al.*, 1997]), which is part of the "Aerosol, Clouds and Trace gases Research Infrastructure" (ACTRIS). The station is located on the northeastern coast of Crete on top of a hill, facing the Mediterranean Sea within a sector from 270° to 90°. It is surrounded by sparse vegetation and olive tree plantations, without significant human activity in the near vicinity, except a small village 3 km to the South. Heraklion, the closest, more densely populated area (~200'000 inhabitants), is situated approximately 50 km west of Finokalia.

Operational meteorological observations, such as wind speed, wind direction, temperature, relative humidity and solar radiation are available at the station. In addition to classical air quality parameters (ozone, nitrogen oxides, carbon monoxide) the station is equipped with a large suite of aerosol measurements.

The halocarbon observations at Finokalia were complemented with data from the AGAGE sites at Jungfraujoch and Mace Head and from Monte Cimone for this study. The high-altitude site Jungfraujoch (JFJ, 7.99°E, 46.55°N, 3573 m a.s.l.) is located in the northern Swiss Alps. It is usually exposed to free tropospheric air but can also be affected by polluted boundary layer air from both sides of the Alps [*Henne, et al.*, 2010; *Herrmann, et al.*, 2015; *Zellweger, et al.*, 2003]. The Mace Head observatory (MHD, 9.90°W, 53.33°N, 15 m a.s.l.) on the west coast of Ireland is normally exposed to relatively clean air from the North Atlantic Ocean but can also be influenced by continental European air masses under certain atmospheric transport conditions. Similar to Jungfraujoch the high-altitude site Monte Cimone (CMN, 10.70°E, 44.18°N, 2165 m a.s.l.) in the Apennine Mountains in Northern Italy is often situated in the lower free troposphere, but especially during daytime receives polluted boundary layer air [*Bonasoni, et al.*, 2000].

### 2.2      Analytical Methods

In situ measurements of halocarbons at the Finokalia observation site were conducted using a gas chromatograph (Agilent 6890) mass spectrometer (Agilent 5973) (GCMS), coupled to an adsorption desorption system (ADS) for pre-concentration of samples from the air [*Simmonds, et al.*, 1995]. A similar instrument with a nearly identical air handling system is used at Monte

Cimone [*Maione, et al.*, 2013]. The ADS is the predecessor of the Medusa pre-concentration unit, which is currently used at the AGAGE sites Jungfraujoch and Mace Head [*Miller, et al.*, 2008].

Two litres of air were sampled every 2 hours, with a collection duration of 40 min, 2 m above the rooftop of the station building, using an inlet facing the open sea. For the correction of short-term drifts of the mass spectrometer response, a working standard was measured after each 10th air sample analysis. Two such standards were used throughout the project, both real-air samples compressed into internally electro-polished 34 L stainless steel canisters (Essex Cryogenics,

Missouri, USA) at Rigi-Seebodenalp (Switzerland), using an oil-free diving compressor. These working standards were calibrated against standards provided by the Scripps Institution of Oceanography (SIO). All results are reported on SIO calibration scales and expressed as dry air mole fractions in parts per trillion (ppt), $10^{-12}$. The respective scales are SIO-05 for HFC-134a, HFC-152a, HCFC-22 and HCFC-142b, SIO-07 for HFC-143a and SIO-14 for HFC-125.

The measurement precision, which is calculated separately for each compound, was estimated as the standard deviation of the working standard observations, inside a moving window covering 10 standard measurements (SI-Table 1). Note that the precision for the ADS measurements at Finokalia was up to an order of magnitude worse than for the sites equipped with the Medusa system. This was partly caused by less frequent reference gas measurements by the

ADS compared to the Medusa. Nevertheless, for the atmospheric inversion this reduction in measurement precision can be tolerated, since the largest part of the total uncertainty in the inversion is contributed by uncertainties in the transport model.

## 2.3     Data Treatment

Data quality was ensured by examining chromatographic quality and comparing observed mole

fractions to observations at selected European AGAGE sites (JFJ, MHD, CMN). Specific observations, showing poor chromatographic quality or unrealistic measurement behaviour were excluded from the time series.

Due to hardware problems of our mass spectrometer, no measurements were conducted from 22 March to 14 April. During the summer (June to August), the observation data behaviour of

HFC-134a and HFC-125 suggested a local pollution source in the vicinity (a few 100 m) of the station, assumed to be a leaking refrigeration/air conditioning system close by. Because the transport model (see Section 2.4) cannot account for such local emissions, HFC-125 and HFC-

134a data were removed during the summer when local wind speeds were below 4 m s$^{-1}$ and the wind direction was north-northeast to east.

Since the transport simulations can only account for the regional emissions in a limited domain and during the time of backward integration, it was necessary to obtain a baseline mole fraction that represents the conditions at the endpoints of the transport simulation. To this end, a statistical method was applied to the observations assuming that a considerable part of the observations was not, or only weakly, influenced by emissions within the period of the transport simu-

lation. The "Robust Estimation of Baseline Signal" (REBS) algorithm [*Ruckstuhl, et al.*, 2012] detects these baseline observations by iteratively fitting a local linear regression model to the data, excluding data points outside a range around the baseline and finally arriving at a smooth baseline curve. The measured dry air mole fraction, $X_O$, can then be represented as the sum of the baseline mole fraction, $X_{O,b}$, and the input due to recent emissions, $X_{O,E}$.

The REBS method was applied separately to the high frequency observation data of each compound and each observation site, using a temporal window width of 30 days and a maximum of 10 iterations with asymmetric robustness weights. Derived mean baseline values for each site and the respective baseline uncertainties, $\sigma_b$, are shown in SI-Table 1. Finally, three-hourly averages were produced from the observations at Finokalia and the other European AGAGE

sites (JFJ, MHD, CMN) in order to match the transport model's temporal output interval.

## 2.4    Transport Simulations

The Lagrangian Particle Dispersion Model FLEXPART (version 9.02) [*Stohl, et al.*, 2005] was used to derive source sensitivities, also referred to as footprints, for 3-hourly intervals at all four observational sites. The source sensitivities quantify the effect of an emission source at a certain

grid location and of unit strength (1 kg s$^{-1}$) on the mole fractions at the receptor. Multiplication of the source sensitivity with an emission field and summation over the entire grid yields the simulated mole fraction at the receptor [*Seibert and Frank*, 2004; *Stohl, et al.*, 2009]. FLEXPART calculates transport by mean and turbulent flow as well as transport within convective clouds. Here, it was driven by meteorological fields obtained from the operational anal-

ysis of the Integrated Forecast System (IFS), provided by the European Centre for Medium-Range Weather Forecasts (ECMWF). Input fields were available at 3-hourly intervals at a global resolution of 1° by 1° and a nested domain with a resolution of 0.2° by 0.2° for the Alpine area. FLEXPART was run in "backward" mode, where 50'000 particles were released from each observation site in 3-hourly intervals and followed 10 days backward in time. Assuming

that emissions are predominantly originating at the ground, the source sensitivities were calculated for a layer reaching from 0 m to 100m above ground. According to the experience of previous studies, the release height of particles, followed by FLEXPART along backward trajectories, was set to 3000 m a.s.l. and 2000 m a.s.l. for the high-altitude stations JFJ and CMN, respectively, where model and real topography differ significantly [*Keller, et al.*, 2012]. For Finokalia, a particle release height of 150 m a.s.l., corresponding to 30 m above the model topography, was chosen, 70 m below the real altitude. However, a comparison between this release height and a release at the true altitude above sea level did not show any significant differences.

Because of the long lifetime of the substances analysed in this study, removal processes were neglected in the FLEXPART simulations. Of the analysed compounds, HFC-152a has the shortest tropospheric lifetime of 1.6 years [*Carpenter and Reimann*, 2014]. Applying this average lifetime, only about 1.7 % of fresh HFC-152a emissions would on average be degraded during the 10-day transport period, whereas typical losses may be larger in summer, but will generally remain smaller than transport uncertainties.

## 2.5    Atmospheric Inversion

To estimate spatially resolved emissions, a Bayesian inversion method [*Enting*, 2002] as implemented and described in *Henne et al.* [2016] was used. Here we only describe the most integral parts of the method and modifications as compared with *Henne et al.* [2016].

In short, the source sensitivities simulated by FLEXPART provide the link to describe a linear relationship between simulated mole fractions at the observation sites, $y$, and an emission field, $x$, which can be written in matrix notation as

$$y = Mx, \tag{1}$$

where $M$ is the source sensitivity matrix constructed from the individual source sensitivities. The state vector, $x$, contains the emissions of each grid cell in the inversion grid and baseline mole fractions, given at baseline nodes at discrete time intervals for each site. Consequently, the matrix $M$ contains two block matrices $M^E$ and $M^B$, denoting the dependence on emissions and baseline mole fractions, respectively. $M^B$ is designed such that the elements represent temporally linear interpolated values between neighbouring baseline nodes [*Henne, et al.*, 2016; *Stohl, et al.*, 2009].

In the Bayesian approach, the a posteriori state, $x_{post}$, is obtained such that the simulations optimally fit the observations, $y_O$, under the presumption of a given prior state $x_{prior}$. This can be achieved by the minimization of the following cost function,

$$J = \frac{1}{2}\left(x_{post} - x_{prior}\right)^T \boldsymbol{B}^{-1}\left(x_{post} - x_{prior}\right)$$
$$+ \frac{1}{2}\left(\boldsymbol{M}x_{post} - y_0\right)\boldsymbol{R}^{-1}\left(\boldsymbol{M}x_{post} - y_0\right),$$

(2)

where the first term gives the deviation of the posterior state vector $x_{post}$ from the a priori state vector $x_{prior}$ and the second term, the misfit between the simulated mole fractions, $\boldsymbol{M}x_{post}$, and the observations, $y_O$. $\boldsymbol{B}$ is the uncertainty covariance matrix of the a priori state vector and $\boldsymbol{R}$ denotes the uncertainty covariance matrix of the data-mismatch and contains both observation and model uncertainties. Section *Covariance Treatment* details how $\boldsymbol{B}$ and $\boldsymbol{R}$ were set up for this study. The diagonal elements of the uncertainty covariance matrix are hereinafter referred to as "analytic uncertainty".

To increase the spatial coverage of our analysis and thereby reduce the uncertainties at the periphery of the Eastern Mediterranean, simultaneous measurements from the three AGAGE sites in Western Europe were included in addition to those at Finokalia. Thus, our inversion grid covered most of Southern and Central Europe, reaching from the Atlantic to the Middle East. To represent the large variety of advection patterns, influencing the observations at the AGAGE sites in our study area, measurements from Dec. 2012 – Dec. 2013 were used in the inversion.

The applied inversion derives spatially-resolved, but temporally-constant emissions. In order to reduce the size of the inverse problem, which depends on the number of grid cells, an inversion grid with variable grid resolution was defined. Grid cells, for which the average source sensitivity was below a predefined threshold were joined with their neighbours until the combined source sensitivity was sufficiently large or up to a maximum horizontal grid size of 6.4° by 6.4°. In contrast to previous studies, using variable grid resolutions [*Brunner, et al.*, 2012; *Henne, et al.*, 2016; *Stohl, et al.*, 2009], the initially computed irregular grid was manually adjusted to ensure that large grid cells did not overlap with different emission regions. This assured a more accurate assignment of emissions per region and their uncertainties, especially in the case of large emissions close to regional borders, and when different a priori uncertainties were given to neighbouring regions.

## 2.6    A priori Emissions

A Bayesian inversion requires a priori knowledge of the state vector to guide the optimisation process. In order to specify a priori emissions and their uncertainty for each grid cell of the inversion grid, emission information was collected on the country/region level and then spatially disaggregated following population density. Since optimising emissions from small and distant (from the observation locations) countries can be afflicted with large uncertainties, we aggregated country-specific a priori information to larger regions (see Table 3 and SI-Figure 2). These were introduced with the intention to separate developed (Annex I/ Non Article 5) and developing (Non Annex I/ Article 5) countries wherever possible. Total a priori uncertainties were assigned to each country/region and each compound separately and then spatially disaggregated following the same population density as for the emissions, which results in constant relative uncertainties for each country/region. This is an improvement as compared with previous studies that used uniform relative uncertainty in the whole inversion domain. [e.g., *Keller, et al.*, 2012].

Our a priori country total HFC emissions for Annex I parties were based on the 2016 National Inventory Submissions to the UNFCCC [*UNFCCC*, 2016] for the year 2013, collected from individual country "common reporting format" tables. To estimate prior emissions for countries within our inversion domain not reporting to the UNFCCC (Non-Annex I), reported emissions were subtracted from estimated global emissions in 2012 provided by [*Carpenter and Reimann*, 2014]. The remaining emissions were further disaggregated to the individual country level, based on population data, provided by the UN population division [*UN*, 2016]. Uncertainties for reported "bottom-up" emissions were arbitrarily set to 20%, whereas estimated a priori emissions for non-reporting countries were given a higher uncertainty of 100% (Table 3). The sensitivity of our posterior emissions to these choices was analysed in additional inversion runs (see Section 2.8).

HCFC-22 global emission estimates provided by [*Carpenter and Reimann*, 2014] were distributed based on regionally estimated shares by *Saikawa et al.* [2012], assuming that contribution ratios of the regions defined in their study have not changed significantly since the period of 2005 – 2009. Emission estimates in areas with differing regional extents in our study compared to that of *Saikawa et al.* [2012] were rearranged using population data. The resulting prior emissions for the European domain compare well with estimated European emissions, derived by *Keller et al.* [2012] during their campaign in 2011. Uncertainties were calculated to add up to

a combined uncertainty of the used global estimate from [*Carpenter and Reimann*, 2014] and the regional estimates derived by *Saikawa et al.* [2012] (Table 5.5).

Based on the assumption that HCFC-142b and HCFC-22 emissions are largely collocated, the same above mentioned regional emission shares are used to derive HCFC-142b prior emissions. Resulting European emissions were further scaled to match HCFC-142b estimates from *Keller et al.* [2012], while Russian emissions, which were not covered in the above mentioned study, were scaled using temporally extrapolated emissions from EDGAR v4.2 [*JRC/PBL*, 2009]. Due to the lack of information and on the basis that Article 5 countries are still allowed to use HCFCs after the phase-out of HCFCs in non-Article 5 countries, North African and Middle Eastern countries within our domain were left unscaled, but given a regional total uncertainty of 100% allowing for substantial corrections of the a priori emissions by the inversion. European regions containing developing and developed countries, as well as Russia were assigned a smaller uncertainty of 50%, reflecting the availability of scaling information.

## 2.7    Covariance Treatment

We followed three different strategies concerning the design of covariance matrices $\boldsymbol{B}$ and $\boldsymbol{R}$. The first two ('Global' and 'Local') use complete uncertainty covariance matrices and are similar to the one used in *Henne et al.* [2016], whereas the third method ('Stohl') assumes uncorrelated uncertainties and uses diagonal-only uncertainty covariance matrices [*Stohl, et al.*, 2009]. The latter has already been used successfully to derive regional halocarbon emissions [e.g. *Keller, et al.*, 2012; *Vollmer, et al.*, 2009].

The uncertainty covariance matrix $\boldsymbol{B}$ of the a priori state-vector consists of two symmetric block matrices $\boldsymbol{B}^E$ and $\boldsymbol{B}^B$, containing the uncertainty covariance of the gridded a priori emissions and the baseline mole fractions, respectively. Diagonal elements of $\boldsymbol{B}^E$, defining the uncertainty of each grid cell emission, were set proportional to the a priori emissions in each cell. The diagonal elements of $\boldsymbol{B}^B$ were set to the constant value of the baseline uncertainty $\sigma_b$, as estimated by the REBS method for each observation site (see Section 2.3), scaled by a constant factor $f_b$. For the covariance methods 'Global' and 'Local' the off-diagonal elements of $\boldsymbol{B}^E$ were defined according to a spatial correlation, decaying exponentially with the distance between a grid cell pair and utilising a correlation length, $L$, which was set to 200 km for all inversions. Furthermore, the baseline mole fractions were assumed to be correlated temporally, described by an exponentially decaying relationship in the off-diagonal elements of $\boldsymbol{B}^B$, based on the temporal correlation length, $\tau_b$, set to 5 days. The choices of the spatio-temporal

correlation lengths did not largely impact the regional emission estimates, when varied within a reasonable range (100 – 500 km for $L$ and 2 – 14 days for $\tau_b$. The choices are based on values estimated in previous studies [*Brunner, et al.*, 2012; *Henne, et al.*, 2016], where maximum likelihood optimisation was used to establish these covariance parameters. For the covariance method 'Stohl', $\boldsymbol{B}$ only contained values in the diagonal, implying uncorrelated a priori uncertainties. For all three approaches, it was assured that the total by-region a priori uncertainty of emissions is the same as defined above.

The covariance matrix $\boldsymbol{R}$ contains the uncertainty of the observations and the model (data-mismatch), $\sigma_c = \sqrt{\sigma_O^2 + \sigma_{model}^2}$. For the covariance methods 'Global' and 'Local' the diagonal elements of $\boldsymbol{R}$ were defined as a combination of the observation uncertainty σ_O and the model uncertainties $\sigma_{model}$. $\sigma_O$ contained the measurement uncertainty (see Section 2.2) and $\sigma_{model}$ was calculated iteratively for each site, incorporating the root mean square error (RMSE) between simulation and the observed mole fractions. The iteration included the use of a posteriori residuals from the previous iteration and follows the description in Stohl et al. [2009]. Off-diagonal elements of $\boldsymbol{R}$ were assumed to follow an exponentially decaying structure [*Henne, et al.*, 2016]. The temporal correlation length, $\tau_c$, of the combined uncertainty, $\sigma_c$, was based on the autocorrelation of the a priori model residuals. Two different approaches were followed to determine $\tau_c$. First (method 'Global'), a constant value of $\tau_c$ for the entire time period and each site was estimated fitting an exponential decay to the first two lags of the global autocorrelation function of the residuals. In a second approach ('Local'), the autocorrelation was evaluated locally within moving windows with a half-width of 80 data points (10 days). Again, $\tau_c$ was then calculated from an exponential fit to the first three values of the autocorrelation function for each window. These procedures to estimate $\tau_c$ worked successfully for all compounds and sites, except for HFC-143a at Finokalia, for which large, unexplained peaks in the observed time series lead to very large values in the autocorrelation function and consequently $\tau_c$. To allow for a meaningful inverse adjustment, a constant $\tau_c$ was used for HFC-143a, based on the mean value of $\tau_c$ for the other compounds.

In the alternative approach ('Stohl') $\boldsymbol{R}$ was specified similar to the above-mentioned method, using the RMSE between a priori simulation and observations. In addition, the extreme values in the residual distribution were filtered and assigned larger uncertainties, in order to derive a more Gaussian distribution of the a priori residuals normalised by $\sigma_c$ [*Stohl, et al.*, 2009]. As a result, a disproportional influence of extreme values, which were not resolved well by the

transport model, can be avoided. Furthermore, off-diagonal elements in **R** were set to zero in this approach.

## 2.8 Sensitivity Inversions

The a posteriori uncertainty, analytically estimated by a Bayesian inversion, often strongly depends on assumptions made on the a priori and data-mismatch uncertainty as well as on the general design of the inversion system. A number of previous studies have shown that this analytical uncertainty is often too small to realistically cover the real a posteriori uncertainty [e.g., *Bergamaschi, et al.*, 2015]. To further explore the range of this structural uncertainty of the inversion setup and test the robustness of the a posteriori results, a set of sensitivity inversions were performed (Table 1).

The inversion using the a priori emissions as described above, the 'Global' method for setting up the covariance matrices **B** and **R**, and observations from all four sites was chosen to represent the base inversion (BASE) setup. The base case does not necessarily offer the best inversion settings for each substance and each site, as these are generally not known, but serves as a starting point to assess the sensitivity of the inversion towards differently chosen parameters.

A first set of sensitivity inversions was used to analyse the effect of different covariance matrix designs. In contrast to the BASE inversion, S-ML and S-MS used the 'Local' and 'Stohl' approach as described in Section 2.7.

We then explored the sensitivity of our a posteriori results towards a priori emission uncertainties, with regard to the inhomogeneous availability of a priori information on halocarbon emissions within our inversion grid. To this end, the a priori uncertainty for each region was increased/decreased by 50 % as compared to the base uncertainty (S-UH, S-UL). Furthermore, two sensitivity runs with 30 % lower and 30 % higher a priori emissions than our BASE inversion, but with the same relative spatial distribution, were conducted (S-PL, S-PH).

In a third set of sensitivity runs the influence of the additional observations gathered during the campaign at Finokalia on the a posteriori emissions in western Europe, central Europe and the Eastern Mediterranean was tested. One sensitivity inversion was set up, excluding the observations from Finokalia (S-NFKL), whereas in a second inversion, only measurements from Finokalia were taken into account (S-OFKL). Using this approach, two questions can be answered. First, what is the gain of the Finokalia observations for top-down emission estimation in the Eastern Mediterranean and, second, did the inclusion of the additional AGAGE sites

provide substantial constraints for the same area? However, the results from these inversions were not added to our overall emission estimate, since they only serve to highlight the importance of a denser observational network.

A final area of structural uncertainty, the baseline assignment, was not further explored in this study. Depending on the setup the definition of the baseline and its treatment in the inversion can have considerable impacts onto the a posteriori results [*Brunner, et al.*, 2017], especially for compounds with small excursions from a variable background such as $CH_4$ [*Henne, et al.*, 2016]. In the case of HFCs and HCFCs the temporal baseline variability is generally small and the pollution peaks are comparably high, somewhat reducing the uncertainty associated with the baseline estimate. Hence, we did not explore this source of uncertainty in more detail in the present study.

## 3   Results and Discussion

In this section, an overview about the measurements taken in Finokalia (FKL) is followed by a comprehensive presentation and discussion of the inversion results. The performance of the BASE inversion is shown exemplarily for HFC-134a in more detail before the results of the sensitivity inversions are presented, highlighting the differences between the BASE case and these inversions. The "top-down" emission estimates for defined regions within the inversion domain are shown in Section 3.4 and are summarised in Section 3.5. The discussion concludes with an additional analysis of seasonality and the benefits of additional measurement sites (Sections 3.6 and 3.7).

### 3.1   Flow Regime and Observations at Finokalia

During our measurement campaign from December 2012 to August 2013, local wind observations showed a transition from a northerly wind regime in December to a more variable wind regime with a bias towards westerly directions from January to June. July and August were characterized by very constant easterly to north-easterly winds. These local observations agree with the results of the atmospheric transport simulations, showing air transported to the station from the African continent and the Western Mediterranean in February and March (Figure 2a). The area of influence changes more towards South-eastern Europe in early summer, whereas in July and August, air is transported from a narrowly defined north-easterly sector (Figure 2b).

These conditions observed during the campaign in 2012/2013 agree with previous descriptions of the wind climatology at FKL that also observed two distinct meteorological regimes in Crete.

During the dry season from May to September, air masses are usually advected from central and eastern Europe and the Balkans, whereas the wet season from October to April is more variable in terms of air transport and favours air masses from the African continent and from marine influenced westerly sectors [*Gerasopoulos, et al.*, 2005; *Kouvarakis, et al.*, 2000]. Therefore, the halocarbon observations presented here can be expected to be the result of typical

advection conditions at FKL.

The halocarbon observations collected at FKL during the campaign are shown in Figure 3, together with data from JFJ and CMN for comparison. The range of the observations at FKL and the temporal evolution of the atmospheric baseline signals agreed well between the sites.

For HFC-134a, which is mainly used as a refrigerant in mobile air conditioning and HFC-125,

which is mainly used in residential and commercial air conditioning, the maximum measured mole fractions and the variability at FKL was smaller than what was simultaneously measured at the two other stations. This could be expected from the maritime influence at FKL, with the closest larger metropolitan areas at a distance of 350-700 km, as compared to nearby emission hot-spots for JFJ and CMN (e.g., Po Valley). For HFC-143a, pollution peaks were comparable

to the measurements at CMN during a short period in the beginning of the campaign (Dec - Feb). After this period, the variability decreased with no more large pollution peaks observed. HFC-152a and HCFC-22 observations showed a similar pattern at FKL as at the other sites. Particularly high mole fractions during several pollution periods were observed for HCFC-22, indicating the proximity of emissions possibly from Article 5 countries where the use of HCFCs

has just recently been capped. Although the highest-observed mole fractions were relatively large, they occurred less frequently than those observed at JFJ and CMN. This was probably due to distant but strong pollution sources influencing the observations at FKL. HCFC-142b mole fractions showed large variability and comparably large peak mole fractions during the summer period at FKL, but again with a slightly lower frequency than at JFJ.

The mean baseline values at FKL for HFC-134a, HFC-125, HCFC-22 and HCFC-142b, calculated with the REBS method [*Ruckstuhl, et al.*, 2012], were within a range of ±7 % of the baseline values derived for the other three sites (see SI-Table 1). Maximum baseline deviations of ±13 % were estimated for HFC-143a and HFC-152a as compared with JFJ.

To illustrate the temporal variability of the observations on a shorter time scale a shorter period

(June 2013) is depicted in SI Figure 3. The time series indicates that pollution events at FKL and CMN persisted over several days, whereas at JFJ pollution peaks were more isolated and

probably associated with individual transport events from the atmospheric boundary layer. Furthermore, some of the compounds showed strong correlations at individual sites (e.g., HFC-134a and HFC-125 at CMN), whereas other compounds showed a more isolated behaviour (e.g., HFC-152a at FKL). This already hints towards common source processes in the former case and separate origins in the latter. The special case of HFC-152a in the Eastern Mediterranean will be analysed further in Section 3.6.

## 3.2    Base Inversion

For the BASE inversion, the covariance design based on the 'Global' autocorrelation function, as described in Section 2.7, was used, combined with the complete set of observations from all four sites, including the observations from FKL. Exemplarily, a comparison of simulated prior and posterior HFC-134a with the underlying observations is shown in Figure 4. At all four sites, the simulated a priori mole fractions reproduced the variability of the observations, indicating satisfactory performance of the transport model (see Table 2). Simulations of the a priori mole fractions showed a tendency to underestimate the observations during peak periods at JFJ, MHD, and CMN, whereas the a priori simulation generally overestimated the observations at FKL. Here, a similar behaviour of the a priori simulations was also observed for HFC-152a, whereas the tendency to underestimate the observations (like at the AGAGE sites) was apparent for all other analysed compounds. Since FKL and the AGAGE sites are mostly sensitive to distinctly different regions, the general overestimation in the prior simulations already points towards generally overestimated or spatially misallocated a priori emissions in the Eastern Mediterranean.

For all four stations, the inversion considerably improved the correlation between observations and simulations, which was evaluated based on the coefficient of determination $R^2$ (Table 2). The performance of the simulated a posteriori signal increased to $R^2 = 0.74$ for FKL and MHD, 0.5 for JFJ and 0.54 for CMN, which corresponds to an improvement of $R^2$ by $\Delta R^2 = 0.33$ for FKL, $\Delta R^2 = 0.13$ for MHD, $\Delta R^2 = 0.17$ for JFJ and $\Delta R^2 = 0.15$ for CMN (Table 2). Only accounting for the simulated and observed signal above the baseline, the performance was lower for FKL ($R^2 = 0.29$), JFJ ($R^2 = 0.34$) and CMN ($R^2 = 0.28$). The correlation of the signal above the baseline for MHD ($R^2_{abg} = 0.73$) remains as high as for the complete signal. We can compare our a posteriori coefficients of determination above the baseline ($R^2_{abg}$, Table 2) with previous inversion studies for similar compounds using the same transport model and observations at the sites JFJ and MHD [*Brunner, et al.*, 2012; *Stohl, et al.*, 2009]. For the site MHD, our a posteriori values for $R^2_{abg}$ are very similar to those previously reported, whereas for JFJ our model

performance lies in the middle of reported values for this site and the compounds HFC-134a and HFC-125. Note that the a posteriori model performance alone is not necessarily a good indicator of reasonable inversion results. The performance ranking between the sites and the large above baseline correlation at MHD also agree with our expectations. The latter is due to the coastal location of MHD with negligible emissions west of the site for several thousand

kilometres across the Atlantic Ocean and the fact that synoptic scale flow, which is well captured by the transport model, intermittently drives European emissions towards the site. In contrast, transport to JFJ and CMN is driven by small scale flow systems and baseline conditions are generally less well-defined in free tropospheric conditions that tend to be more variable. Finally, while FKL is a coastal site like MHD, it does not exhibit a well-defined baseline sector,

since emission sources may be found at the entire coastline in the Eastern Mediterranean at distances around 1000 km from the site.

To evaluate the ability of the model to simulate the observed amplitudes correctly, we used the Taylor skill score (TSS), combining correlation and variability of observed and simulated mole fractions [*Taylor*, 2001]. The maximum attainable Pearson correlation coefficient, indicating a

"perfect" simulation in terms of the strength of the relationship between simulated values and observations, was set to 0.9. Thus, a TSS of 1 indicates a perfect simulation with regards to amplitude and correlation, whereas a TSS of 0.65 means that the observed variability is under/overestimated by a factor of 2 for perfectly correlated simulations. Although the normalized standard deviation decreased for FKL, the TSS was increased to 0.95 due to the improvement

of the correlation of posterior results and observations, indicating, that although the relationship of observations and simulations was increased, the inversion did not adjust the amplitudes of the pollution peaks. At CMN the a posteriori TSS increased to 0.74, driven by both an increase of the normalised standard deviation and correlation, whereas the TSS for JFJ and MHD decreased to 0.71 and 0.75 respectively. The latter is due to a reduction of simulated peak heights

compared to the a priori simulation, while the correlation was strongly improved. In general, the resulting Taylor skill scores were in a similar range as in previous regional-scale inversion studies [*Brunner, et al.*, 2017; *Henne, et al.*, 2016].

Model and inversion performance were also evaluated using the Root-Mean-Square-Error (RMSE; a combined measure of variability and bias) between simulated and observed mole

fractions. Its reduction from a priori to a posteriori simulations amounted to 20%, 12% and 10% for JFJ, MHD and CMN, respectively. The absolute a posteriori RMSE was in the range of 2.9 – 5 ppt for these sites. The RMSE improvement for FKL from the a priori RMSE (4.7 ppt) to

the a posteriori RMSE (1.7 ppt) was much larger (64%). This can be attributed to the above-mentioned overestimation of the simulated prior values and the optimisation by the inversion, which also included a considerable reduction of the baseline. Again, these RMSE reductions were in a similar range as those reported in previous studies [*Keller, et al.*, 2012; *Stohl, et al.*, 2009; *Vollmer, et al.*, 2009].

The inversion performance of HFC-125 and HCFC-142b was similar to HFC-134a, with mean posterior TSS of 0.81 and 0.78, respectively, compared to 0.78 for HFC-134a. For HFC-152a, HFC-143a and HCFC-22 they decreased to 0.73, 0.74 and 0.75, respectively (Table 2).

For the BASE inversion of the exemplary compound HFC-134a a posteriori were mostly smaller than a priori emissions with the exception of areas in Northern Italy, Slovenia, Croatia and along the western part of the British Channel (Figure 5). Most pronounced emission differences in the Eastern Mediterranean were associated with the larger urban centres in Greece and Turkey (Athens, Thessaloniki, Istanbul), whereas in Western and Central Europe similarly large reductions were assigned to the Benelux area and the western part of Germany as well as to the UK. Within the same BASE inversion of HFC-134a the analytic uncertainty in the Eastern Mediterranean was reduced by more than 80% from its prior value for grid cells containing large metropolitan areas such as Athens and even Cairo (Figure 5). For Western Turkey and large parts of the Balkans, the uncertainty was reduced by 30-60%. Similar reductions are also achieved over large parts of Western and Central Europe, to which the AGAGE sites are sensitive. Although other adjacent areas such as Middle Eastern countries bordering the Mediterranean Sea (e.g., Israel, Jordan) and countries further Northeast (e.g., Ukraine) were detected during our measurement campaign, the uncertainty was reduced less by the inversion (10-30%).

Similar patterns of uncertainty reduction resulted for HFC-152a, HFC-125, HCFC-22 and HCFC-142b. For HCFC-142b, the reduction was lower for the Balkans (~ 10%), but similarly large for Western Turkey (20-40%). For HFC-143a, the uncertainty was reduced by 20 % for the area of Athens, whereas only negligible reductions were estimated for Turkey.

## 3.3 Sensitivity Inversions

### 3.3.1 Influence of Covariance Design

The first sensitivity inversion, S-ML, uses the 'Local' approach to estimate the temporal correlation length scale of the data-mismatch uncertainty (see Section 2.7). As a consequence, the weights, different observations were given in the inversion, were redistributed as compared

with the BASE inversion. The total covariance by site that is contained in **R** can be calculated

by

$$\sigma_{k=}\sqrt{\frac{R_k \cdot R_k^T}{N_k}},$$    (3)

where $\boldsymbol{R_k}$ is the block matrix belonging to all $N_k$ observations/simulations of an individual site.

In the case of our exemplary compound HFC-134a, $\sigma_k$ took values of 4.2, 7.3, 8.7 and 8.7 ppt for our BASE inversion (global $\tau_c$) and the sites FKL, JFJ, MHD and CMN, respectively. For the S-ML sensitivity inversion (local $\tau_c$) these values only differed slightly for the sites FKL

and CMN, but were 8.3 ppt and 9.0 ppt for the sites JFJ and MHD, respectively. As a consequence less (more) weight was given to the observations from JFJ (MHD) in S-ML than in the BASE inversion. Especially for MHD one would thus expect that the a posteriori performance would be increased in the S-ML case compared to the BASE inversion. This was not the case (see below). A possible reason can be found in the distinctly different temporal pattern of the

temporal correlation length scale. The differences between the empirical auto correlation function for a running window width of 10 days (local) and the fitted auto correlation function with a constant (global) correlation length scale for the site MHD is shown in SI Figure S4. MHD infrequently received pollution events from the European continent. These episodes were characterized by relatively large model residuals. Also the auto correlation of the residuals during

these periods was enhanced. The global estimate of $\tau_c$ then lead to an underestimation of auto-correlation during these periods (indicated by positive values in panel d of SI Figure S4). Finally, this means that in the BASE inversion more weight (smaller auto correlation, and, hence, smaller covariance) was given to the observations from MHD during the pollution events as compared to the sensitivity inversion with local $\tau_c$. In turn, the posterior adjustments for MHD

had a larger impact for the BASE inversion and performance improved more than in the S-ML case.

The model performance in terms of the RMSE was similar to the BASE inversion at FKL, CMN and JFJ. For MHD the RMSE was not reduced by the inversion, thus, compared to the base inversion, posterior RMSE values were 14% higher. The same pattern was observed for the

coefficient of determination $R^2$, which was increased by less than 2% for FKL, CMN and JFJ, but dropped by approximately 8% at MHD. Despite the slight increase in the correlation at FKL, CMN and JFJ, the Taylor Skill Score decreased between 1-4%, indicating that in the S-ML case, the peak amplitudes are not as well simulated as in our BASE inversion. For MHD,

the TSS was reduced by 12%, reflecting that in addition to the lower correlation, S-ML also underestimated the peak amplitudes at this coastal location.

The sensitivity case S-MS used uncorrelated a priori and data-mismatch uncertainties (see Section 2.7). As opposed to S-ML, the RMSE of S-MS for HFC-134a was improved by 14%, 6% and 2% at MHD, JFJ and CMN respectively, as compared with the BASE inversion, whereas no improvement was observed for FKL, which showed a small RMSE of 1.7 ppt in the BASE inversion already (SI-Table 2). $R^2$ was generally higher for S-MS compared to the BASE inversion. It increased between 1-3% for FKL, CMN and MHD and by 6% for JFJ, showing the best absolute performance for MHD and FKL in the posterior $R^2$, with 0.76 and 0.75, respectively. As indicated by higher Taylor skill scores (SI-Table 2), S-MS was also able to more closely reproduce the amplitude of the peaks at all sites as compared with the BASE inversion.

Total HFC-134a emissions for the whole inversion domain were 10% lower for the S-ML case, whereas they were 30% higher for S-MS, as compared to the BASE inversion. While regional emissions from Greece and the Balkans were relatively unaffected in the S-ML case, more pronounced negative deviations compared to BASE were established for Turkey (-14%), Central W (FR, LU, NL, BE; -23%) and the Iberian Peninsula (ESP, PT; -22%) (Figure 5 and Figure 6b,c). A posteriori differences were less smooth in the S-MS inversion as compared to the BASE and S-ML inversions (Figure 6), reflecting the effect of not using a spatial correlation in the a priori emissions. Regional emissions estimated with S-MS were generally higher as compared to the base inversion (Figure 6c). Significantly (40%, $p < 0.05$) higher emissions were obtained in the UK and Ireland compared to the BASE inversion. Regional emissions of North-Western Europe and the Balkans were larger by 20-60% in S-MS. Note that in our S-MS inversion both covariance matrices didn't contain off-diagonal elements, whereas both matrices did in the BASE case. Alternatively, it could have been beneficial to isolate the influence of correlated uncertainties in each matrix independently, i.e. use data-mismatch covariance as in S-MS with the a-prior covariance of the BASE case.

In summary, S-ML showed a slightly weaker performance than the BASE inversion, with insignificantly lower total emission estimates but similar analytic uncertainties. On a regional level, the impact of S-ML on the estimated emissions varies by region, showing less influence on the Balkans and Central W, whereas larger deviations were seen for Turkey, Western and the British Isles. On the contrary, S-MS performed slightly better and resulted in generally larger emissions than the BASE inversion, but confirmed the significant emission reductions as compared to the a priori emissions.

### 3.3.2  Influence of A Priori Uncertainty

To assess the influence of our regionally assigned a priori uncertainties, the sensitivity inversions S-UL and S-UH were run with 50% smaller and larger a priori emission uncertainties as compared to the base inversion. As expected, a posteriori model performance generally increased with larger a priori uncertainties because the optimisation is less constrained by the prior. However, HFC-134a domain total a posteriori emissions remained similar to those in the BASE inversion, whereas S-UL resulted in slightly increased emission estimates, remaining closer to the prior emissions (see supplement). A posteriori HFC-134a emission uncertainties were decreased (increased) by ~28% and ~16% in comparison to the BASE inversion, if a priori emission uncertainties were smaller and larger, respectively (SI-Figure 3).

In general, the absolute emission estimates for the study domain seemed to be very robust to changes in the a priori uncertainty. A posteriori emission estimates for the case with lower a priori uncertainties (S-UL), comprising all the analysed species except HFC-134a, showed insignificantly larger total emissions. This reflects the constraint, which requires the results to follow the a priori emissions more closely in this case. Total a posteriori emissions in the case of larger a priori emission uncertainties remained close to our BASE case. Emission uncertainties in the a posteriori, as compared to the BASE inversion, were on average about 18% higher and 27 % lower for the S-UH case and the S-UL case, respectively. This tendency can be expected from the a priori emission uncertainties. The results of these two sensitivity inversions emphasize the general robustness of the inversion system to changes in the a priori emission uncertainties. Exceptions in the case of HFC-134a are discussed in Section 3.4.

### 3.3.3  Influence of Absolute A Priori Emissions

In order to assess the sensitivity of the results on the absolute magnitude of the a priori emissions, we performed additional sensitivity inversions with 30 % lower and 30 % higher a priori emissions compared to our BASE case (S-PL, S-PH). Even for the low a priori, a-posteriori emissions were smaller for most compounds and regions. However, we could not observe a strong influence of the total a priori emissions onto the a posteriori emissions. As an indicator, the ratios between the a priori and a posteriori emissions were calculated for the sensitivity inversions using high and low a priori emissions. The ratio was 1.85 for the a priori emissions, as prescribed by the input, whereas it ranged from 0.93 to 1.25 for the a posteriori emissions and most regions and compounds. Consequently, in most cases the range in a posteriori emissions spanned by these variations in the a priori was smaller than the analytic uncertainties of

the a posteriori emissions. Exceptions to this reduction in the ratio between high and low a posteriori emissions were HFC-152a emissions from Greece (a posteriori ratio of 2.4). In this case a posteriori emissions were significantly larger for the high a priori inversion than for the base and low a priori inversion. Furthermore, the ratio only slightly decreased for HCFC-142b emissions from Greece and Turkey (a posteriori ratio of 1.6). However, in the latter case the a posteriori uncertainties were still larger than the range of these sensitivity runs. This clearly indicates that especially for the well simulated species the dependency on the prior emission level is not the main source of uncertainty of the a posteriori emissions.

### 3.3.4 Seasonality of HFC-134a emissions

A number of authors have suggested increased emissions of halocarbons used as refrigerants during the warm season [e.g., *Hu, et al.*, 2017; *Xiang, et al.*, 2014] due to the more frequent use of refrigeration and air conditioning applications. In general, we did not focus on the seasonality of the emissions because our observations in the Eastern Mediterranean did not cover a complete annual cycle and, therefore, temporally variable a posteriori emission estimates may suffer from this lack of observations. The latter is especially true since we also observed seasonally variable main advection directions at the site. However, we performed one additional inversion with seasonally variable emissions of the widely used refrigerant HFC-134a (the most abundant and best simulated compound). As expected, we find mixed results for the Eastern Mediterranean, where for Greece and Turkey the maximum a posteriori emissions were derived for the fall (SON), not the summer (JJA) (see SI Figure 6). However, this is mainly due to the lack of observations in this period and the a posteriori staying close to the a priori. The emission totals for both countries were considerably higher when seasonality was considered. However, this can mainly be explained by the higher and not well constrained SON emissions. Without a complete year of observations in this area, it is impossible to finally assess the consequences of the assumption of temporally constant fluxes that was used in all other inversions in this work. In Western Europe we observed a clear seasonality with elevated summer emissions for Italy (+90 % above winter emissions), Germany (Central W, +85%), the Iberian Peninsula (+135 %) and the British Iles (+115 %), but not for France and the Benelux region (-22 %). These variable results for Central Europe indicate the increased uncertainties that result from the reduced number of observations to constrain each individual flux. Our estimates were on the order that was previously reported on a global scale for HFC-134a emissions [*Xiang, et al.*, 2014], but were considerably larger than the 20-50 % summer time increase estimated for HFC-134a in USA [*Hu, et al.*, 2015] and HCFC-22 in Western Europe [*Graziosi, et al.*, 2015]. Slightly larger

seasonal amplitudes (1.5-2) were reported in an updated, more recent study for a number of HFCs and HCFCs in the USA [*Hu, et al.*, 2017]. Total annual emissions in the regions experiencing a seasonal cycle were slightly enhanced compared to our BASE scenario but remained well within the reported a-posteriori uncertainties. From these comparisons we conclude that neglecting seasonality in the inversion may introduce a small negative bias in our a posteriori estimates, but that at least for HFC-134a this bias falls within our uncertainty estimate.

## 3.4 Regional Total Emissions

Our estimated regional total emissions are summarized in Table 3 and Figure 7. The "top-down" emission estimates presented here are the mean values of the BASE and six sensitivity inversions (S-ML, S-MS, S-UH, S-UL, S-PH, S-PL). The uncertainty range given here and in Table 3 represents the range of these five inversions based on their mean values and the analytical a posteriori uncertainty (95% confidence interval), whichever is larger. This measure was chosen to accommodate, on the one hand, the analytical uncertainty as estimated by the Bayesian formulation and estimated for each inversion run as the a posteriori uncertainty, and, on the other hand, the structural uncertainty that is reflected by the spread of the sensitivity inversions and results from choices in the parameter selection of the covariance design. The comparison between structural and analytic uncertainties reveals that the dominating type of uncertainty varies largely between different compounds and different regions. For most compounds and regions, the two types of uncertainty fall within a similar range (HFC-152a; HFC-143a; HFC-125; HCFC-22; HFC-134a only in the eastern part of the domain). For HCFC-142b the structural uncertainty was generally smaller than the average a posteriori uncertainty. In contrast, for HFC-134a and the western part of the domain (British Isles, Iberian Peninsula, Western, Central W) the structural uncertainty was clearly larger than the analytical uncertainty.

This relatively large spread in the sensitivity inversions results from the differences between the sensitivity inversions with different covariance matrices (S-ML and S-MS), where a general tendency to smaller changes from the a priori (resulting in larger a posteriori emission) was observed for the western part of the domain and for Turkey. In addition, a similar tendency was observed for the same regions, except the Western region, when different a priori uncertainties were applied (S-UH, S-UL, SI-Figure 3). Therefore, combining the results from all sensitivity inversions revealed relatively large uncertainties in the "top-down" estimates in a region that is relatively well covered by the existing AGAGE network and emphasizes the use of such sensitivity tests to explore the real uncertainty of the "top-down" process and the need for more objective methods to derive the data-mismatch covariance matrix.

### 3.4.1 HCFCs

HCFC-22 is the most abundant HCFC in today's atmosphere and has been widely used as a refrigerant and foam blowing agent in much larger quantities than other HCFCs. Due to regulations by the Montreal Protocol, global emissions have remained constant since 2007 [*Carpenter and Reimann*, 2014]. Our "top-down" emission estimate for the regions listed in Table 3 (in the following referred to as total emissions) amounted to 9.0 (7.1-10.7) Gg yr$^{-1}$. As expected, high emissions were concentrated in regions defined by the Montreal Protocol as developing (Article 5) countries, such as Egypt, the Middle East and Turkey, accounting for 44% (17-72%) of the total emissions. Our estimates for Central and Western European (regions Western, Central W, British Isles, Iberian Peninsula and Italy) emissions are 3.1 (1.7-4.5) Gg yr$^{-1}$, which is 69% (38-100%) less than reported by *Keller et al.* [2012] for the same area in 2009, which may indicate that HCFC-22 emissions continue to decrease in these developed countries. However, major pollution events were observed at FKL when air arrived from areas such as Egypt, which may be explained by the fact that caps to HCFC production and consumption for Article 5 parties began only in 2013. For the total domain, our a posteriori estimates were significantly lower than the a priori values. On the regional scale, a posteriori estimates were larger than a priori for the above-mentioned Article-5 countries (Egypt, Middle East), whereas this tendency was inversed for Non-Article 5 countries. These results agree with the expectation that due to the stepwise phase out of HCFCs in developing countries and the inherent time lag until release to the atmosphere [*Montzka, et al.*, 2015], HCFC-22 emissions remain at considerably high levels.

HCFC-142b is applied mainly as a foam blowing agent for extruded polystyrene boards and as a replacement for CFC-12 in refrigeration applications [*Derwent, et al.*, 2007]. Our total estimated emissions sum up to 1.0 (0.8-1.2) Gg yr$^{-1}$. Turkey, listed as an Article 5 party, accounts for 13.9% (2.5-25 %) of these total emissions, whereas the contribution of other Article 5 regions is less pronounced as compared to HCFC-22. Average a posteriori emissions in the Eastern Mediterranean (regions Greece, Turkey, Middle East, Egypt, Balkans and Eastern) are estimated to 0.38 (0.00-0.80) Gg yr$^{-1}$, which is 38 % of the domain total emissions. However, our inversion was not able to significantly reduce the uncertainty estimate for these regions, demonstrating the need for additional and continuous halocarbon measurements in this area. HCFC-142b emissions in Central and Western Europe, where the use of HCFCs has practically been phased out, show a comparatively large contribution of 0.53 (0.36-0.70) Gg yr$^{-1}$, which accounts for 52% (35-69%) of the domain total emissions. Although the spatial distribution of

HCFC-142b emissions in Central Europe resembles the pattern derived by *Keller et al.* [2012], dating back to emissions from 2009, our estimates are lower by a factor of ~2. Our estimates are also lower by the same factor ~2 compared to bottom-up estimates of HCFC-142b emissions, as reported in EDGAR v4.2 [*JRC/PBL*, 2009] for the year 2008, for both Western Europe and the Eastern Mediterranean. However, the latter is mainly driven by generally smaller emissions in the Eastern and Balkan regions, whereas for Turkey, the Middle East and Egypt larger than EDGAR v4.2 values were estimated by the inversion. The general decrease within the domain is in line with global emissions of HCFC-142b, which are considerably lower than those of HCFC-22 and have declined by 27%, from 39 (34-44) Gg yr$^{-1}$ to 29 (23-34) Gg yr$^{-1}$ between 2008 and 2012 [*Carpenter and Reimann*, 2014; *Montzka, et al.*, 2015]. The comparison of a priori and a posteriori emissions of HCFC-142b shows a much more diversified pattern than for HCFC-22, with regions such as Turkey and Western E, where our bottom-up assumptions were too low, whereas they were too high for Maghreb and Egypt and agreed well for Italy, Greece and Central W.

### 3.4.2   HFCs

HFC-134a is currently the preferred refrigerant in mobile air conditioning systems and, together with HFC-125, which is mostly used in refrigerant blends for stationary air conditioning  and commercial refrigeration, belongs to the two most popular HFCs in Europe [*O'Doherty, et al.*, 2004; *O'Doherty, et al.*, 2009; *Velders, et al.*, 2009; *Xiang, et al.*, 2014]. This is reflected by the large amplitude and frequency of pollution peaks, which were observed at all continuous observations sites but especially at JFJ and CMN (Figure 3). Total simulated HFC-134a emissions for our analysed regions were 18.6 (16.7-20.6) Gg yr$^{-1}$. Emissions from Eastern Mediterranean (Greece, Turkey, Balkans, Eastern, Middle East, Egypt) summed up to 4.5 (1.7-7.3) Gg yr$^{-1}$, which is ~24% of the domain total emission. Another 63% were emitted from Central and Western Europe, totalling at 11.7 (9.0-15.3) Gg yr$^{-1}$. Comparing the aggregated emissions of reporting regions to UNFCCC inventories reveals that the inversion generally estimated a posteriori emissions of HFC-134a that were 51.4% (36.8-68.7%) lower than the respective UNFCCC reports. Only HFC-134a emissions of Italy and Eastern European countries were within the range of reported UNFCCC estimates. Furthermore, our results suggest lower emission in most region in comparison to EDGAR v4.2_FT2010 [*JRC/PBL*, 2009] for the year 2010, with the exception of Greece, Turkey and the Eastern region, where both estimates are very similar, and of Egypt and the Maghreb region, where the inversely estimated emissions were considerably larger than EDGAR values.

These findings of generally smaller than reported HFC-134a emissions in Western and Central Europe resemble the results of other studies performed for earlier years [*Brunner, et al.*, 2017; *Lunt, et al.*, 2015; *Say, et al.*, 2016]. The differences between the country-wide emissions reported to UNFCCC and the range of results found in this study seem to be somewhat more

pronounced than in previous studies. This is consistent with *Brunner et al.* [2017], who reported a relatively large range of regional emission estimates depending on the employed inverse modelling system.

HFC-125 domain-total emissions were estimated at 8.1 (7.3-8.8) Gg yr$^{-1}$ with emissions from the Eastern Mediterranean contributing 15 % or 1.2 (0.2 – 2.2) Gg yr$^{-1}$. This compares to global

emissions of about 50 Gg yr$^{-1}$ as estimated by global inverse modelling for the period 2011-2015 [*Simmonds, et al.*, 2017]. Our results for Turkey agree well with those reported to UNFCCC, but are three times smaller than EDGAR v4.2 FT2010. For Greece, our estimate of 0.25 (0.17-0.32) Gg yr$^{-1}$ falls between the much larger UNFCCC value of 0.60 Gg yr$^{-1}$ and the smaller EDGAR v4.2 FT2010 estimate of 0.1 Gg yr$^{-1}$. Emissions from the Eastern region, the

Middle East and Egypt remained relatively close to the a priori estimates, whereas for the Balkans we derive a 50 % increase compared to the a priori emissions to 0.18 Gg yr$^{-1}$, which is still considerably smaller than the EDGAR v4.2 FT2010 value of 0.55 Gg yr$^{-1}$. This stands in contrast to the results of *Keller et al.* [2012] for the Eastern region, showing large discrepancies between "top-down" and "bottom-up" estimates in some of these countries, most likely caused

by unrealistically low values reported to UNFCCC. Besides the fact that the estimates of *Keller et al.* [2012] rely on measurements from Hungary, with a better coverage of North-Eastern Europe than we have from FKL, the discrepancies would be smaller in a retrospective view, because HFC-125 bottom-up emissions of several Eastern European countries were revised upward in the 2016 submissions to the UNFCCC for the year 2009. The largest part of the remain-

ing HFC-125 emissions (71 %) was allocated to Central and Western Europe by the inversion, and was about 30 % lower as compared to the a priori estimate with the exception of Italy, where a posteriori values were very close to those reported to UNFCCC. Our results for Western and Central Europe broadly agree with those reported by *Brunner et al.* [2017] and *Lunt et al.* [2015]. However, note that *Brunner et al.* [2017] describes a substantial underreporting of

HFC-125 emission from the Iberian Peninsula in 2011, whereas we find an overestimation by ~25% for 2013. This has to do with a retrospective revision of the Spanish UNFCCC reporting, which resulted in a doubling of most HFC emissions reported in 2016. In absolute terms, our estimates of 1.5 (1.2-1.8) Gg yr$^{-1}$ for the year 2013 agrees well with that given in *Brunner et al.*

[2017] for the year 2011 (1.1 – 2.8 Gg yr$^{-1}$). For Italian HFC-125 emissions our result of 1.05 (0.91 – 1.19) Gg yr$^{-1}$ is at the lower range given by Brunner et al. [2017]. However, note that in their case only one out four inversion systems yielded twice as large a posteriori emissions for Italy, whereas the other systems agreed closely at values around 1 Gg yr$^{-1}$. Also note that one of their inversion systems was the one used here using the diagonal only covariance matrices (S-MS).

HFC-143a is another major HFC, which is commonly used in refrigerant blends for commercial refrigeration. It is sparsely used in Eastern European countries (Balkans, Eastern, Greece and Turkey), where our "top-down" estimate showed combined annual emissions of 0.36 (0.14-0.58) Gg yr$^{-1}$, which corresponds to 6.3% (2.5-10.0%) of the domain total of 5.7 (5.3-6.3) Gg yr$^{-1}$. Emissions higher than the a priori estimates were determined for Maghreb and Egypt with 0.41 (0.15-0.67) Gg yr$^{-1}$ and 0.24 (0.04-0.44) Gg yr$^{-1}$, although relatively large uncertainties are connected with these values, since advection from the respective regions was not often observed. 80% of the HFC-143a emissions within our domain have their origin in Central and Western Europe, with the main sources in the Western region and the Iberian Peninsula. Our estimates agree within 10% with reported UNFCCC values on the domain total basis. For Turkey and the Eastern region, as well as the Iberian Peninsula and the British Isles, reported values agree closely with our estimates ($\Delta$ emission estimates $< 7\%$), whereas our estimates of Central E, Central W, Western, Italy and Greece are 18-35% lower than UNFCCC values.

HFC-152a has the smallest 100-year global warming potential of the major HFCs and is primarily used as foam blowing agent and aerosol propellant. Our domain total "top-down" estimate was 2.8 (2.3-3.3) Gg yr$^{-1}$, which corresponds to only around 6% of estimated global emissions [*Simmonds, et al.*, 2016]. South-eastern Europe's (Greece, Turkey, Balkans and Eastern) annual emissions were estimated at 1.2 (0.6-2.0) Gg yr$^{-1}$, corresponding to 43% (22-74%) of total domain emissions. The largest emissions from any individual region were established for Turkey, 2-3 times higher than our estimates for all other regions within the inversion domain. However, this is still almost a factor of 2 lower than what Turkey reports to the UNFCCC. The UNFCCC inventory of Greece overestimates the posterior emissions inferred in this study by a factor of 5. However, it is known that for the UNFCCC, emissions of HFC-152a are reported in the country where the consumer product is manufactured, not in the country where emissions are occurring during use or disposal. For example, if a foam is blown in country X and sold to country Y, emissions would mainly occur during usage in country Y but are reported under country X. From a global perspective, this makes sense but is not compatible with real

emissions in the respective countries. Emissions from Non-Annex I countries belonging to the Middle East and Northern Africa (Maghreb, Egypt) are small (0.43 (0.00-1.0) Gg yr$^{-1}$). Our "top-down" estimates for Central and Western Europe make up for the remaining 0.87 (0.58-1.20) Gg yr$^{-1}$ of the annual HFC-152a emissions. For all Central and Western European countries, reporting values to UNFCCC, we find a general tendency, that "top-down" emissions are lower than UNFCCC values, with largest discrepancies for the Iberian Peninsula and Central E. For the British Isles, our results are a factor of 2 smaller than the findings of *Lunt et al.* [2015] for the years 2010-2012. In contrast, our estimates for the British Isles agreed within their uncertainties with those reported in Simmonds et al. [2016], which is also true for our estimates for the Central W region and the Iberian Peninsula. In contrast, our "top-down" estimates for Italy are a factor 2 smaller than reported by the latter authors. These results underline the findings of Brunner et al. [2017] that regional inversions for halocarbons suffer from the sparsity of the currently existing observational network. In turn it remains very difficult to derive precise top-down emissions for individual countries and regions.

## 3.5    Summary of Halocarbon Emissions

Our best estimate of domain total halocarbon emissions for 2013 was 82.8 (78.1-92.3) Tg $CO_2$eq yr$^{-1}$ for the four analysed HFCs and 17.9 (14.7-24.4) Tg $CO_2$eq for the two HCFCs. This corresponds to 12.2% (11.5-13.6%) and 2.5% (2.1-3.5%) of global halocarbon emissions [*Carpenter and Reimann*, 2014]. The HFC emissions from the Eastern Mediterranean (Greece, Turkey, Middle East, Egypt, Eastern, and the Balkans) accounted for 13.9 (11.3-19.3) Tg $CO_2$eq yr$^{-1}$ and the HCFC emissions from the same region for 9.5 (6.8-15.1) Tg $CO_2$eq yr$^{-1}$.

As expected, per-capita $CO_2$ equivalent emissions of HFCs vary strongly in the Eastern Mediterranean (Figure 8). For Greece, per capita emissions were similar to other Western European countries, whereas for the developing countries (Article 5 countries) in the Eastern Mediterranean (Turkey, Middle East), with the exception of Egypt, per-capita HFC emissions were much smaller. On the other hand, per capita $CO_2$ equivalents of HCFC emissions were largest in Article 5 countries in the Middle East and Maghreb region, where the phase-out of these compounds is delayed as compared to the Non-Article 5 countries in Western Europe. In this context, it is also interesting to note that the HCFC per-capita emissions from Greece (Non-Article 5) are similarly large as those from its neighbour Turkey (Article-5).

## 3.6 Temporal Variability of HFC-152a Emissions

Some of the larger HFC-152a pollution peaks observed at FKL (see Figure 3) are not well reproduced by the transport model. The atmospheric inversion only slightly improved the comparison, indicating the inability to unambiguously assign an emission region or a constant emission process to these peaks. In the following, the transport situations experienced during the observed HFC-152a peaks are analysed in more detail.

The time series of HFC-152a in FKL (Figure 9c) shows intermittently appearing pollution peaks, most pronounced in June and August, which are badly reflected by the simulations, even when a posteriori emissions are used. Especially two observed broader peaks in June and August are not visible in the simulations. This could be due to inaccuracies in the transport model and weaknesses of the inversion, or because of large, localized, and temporally varying emissions sources, such as HFC-152a production facilities [*Keller, et al.*, 2011]. However, our inversion approach assumes temporally constant emissions and is not able to unambiguously assign a specific source location or area to individually observed pollution peaks that are caused by temporary emissions. For the localization of such emission sources, we used a simple, qualitative approach, by calculating the correlation between the observed HFC-152a time series and FLEXPART simulated source sensitivities in the individual grid cells. First, the correlation for the complete time series was calculated, thereby ignoring the proposed intermittent character of the source. Using this method, generally positive Pearson correlation coefficients were established for all land areas with maximal correlation coefficients located in grid cells in Northwestern Turkey (Figure 9a). To further isolate the potential source areas, correlations were calculated using only peak periods in the observations at FKL, including the times of increasing and decreasing mixing ratios at the flanks of each peak. These results showed a further restriction of significant positive correlation coefficients to Northwestern Turkey, bordering the Marmara Sea and the Bosporus area (Figure 9b), which are both important industrial regions. This result could point to large contributions from the metropolitan area of Istanbul, where HFC-152a could be emitted from installed consumer products. However, due to the strong temporal variability in emissions, which seems to be inherent to the observed peaks, the results are more likely to be explained with large emissions from an industrial facility in the localized regions.

## 3.7 The Impact of Halocarbon Observations at Finokalia

Our campaign in Finokalia added halocarbon observations in an area of Europe from which emissions are only sporadically detected by the existing AGAGE network. We assessed the

added value of a station in FKL, by excluding it from the inversion and estimating Eastern Mediterranean emission only from the existing AGAGE network (S-NFKL). Furthermore, we excluded all stations but FKL from the inversion to test if the existing AGAGE sites add value to our estimate of emissions in the Easter Mediterranean (S-OFKL). The regional emission
estimates using the different station setups are shown in Figure 10. For Greece and Turkey, which were best covered by our observations in FKL, a clear influence of the measurements at FKL on the "top-down" emission estimates can be seen. For HFC-125 and HFC-134a, used as exemplary compounds for this analysis, the inversion excluding FKL was mainly driven by the a priori values, whereas including FKL strongly reduced the emissions and the analytic uncer-
tainty (Figure 11). A similar effect is seen for the Middle East and Egypt, although the number of times during which our site was sensitive to these areas was limited. These results clearly show that regional emission estimates using only AGAGE stations for areas as far as the Eastern Mediterranean are unreliable and an extension of the current network is critical for emission control in this economically very dynamic area.

For Eastern European countries and the Balkan regions, the influence of measurements at FKL reduced HFC-134a and HFC-125 emissions and emission uncertainties slightly. However, Central European measurements have a similar influence on these results. An interesting impact over larger distances can be observed for Italy and the Iberian Peninsula, where the additional measurements from FKL have more of a reducing effect on the absolute emissions than on the
uncertainties, whereas emissions in Central and Western Europe including the British Isles are largely unaffected by our measurements at FKL. The effect of measurements at FKL on modelled emissions from Italy and the Iberian Peninsula can be explained by the additional constraints provided by FKL for Italy. These decreased the estimated Italian emissions and at the same time slightly increased baseline mixing ratios for JFJ and CMN for periods with influence
from the Western Mediterranean. Since simulated source sensitivities are often simultaneously elevated for Italian and Iberian source areas, the increased baseline will translate also to smaller emissions on the Iberian Peninsula even though the observations at FKL were virtually not sensitive to emissions from this region.

The inversion using only observations from FKL (S-NFKL) had virtually no effect on the a
930 posteriori emissions and their uncertainty for Greece, Turkey, the Eastern region and the Middle East as compared with the BASE inversion. For Egypt, the Maghreb countries and the Balkans slightly reduced a posteriori estimates were observed, whereas for Italy, Central and Western Europe the a posteriori estimates differed strongly from the BASE inversion and showed little

uncertainty reduction. These results indicate the importance to include all available halocarbon observations in regional estimates even if these are as distant as Monte Cimone is to Finokalia (~1600 km).

## 4   Conclusion

During a period of six months, from December 2012 to August 2013, we performed continuous halocarbon observations at the atmospheric observation site of Finokalia (Crete, GR) - the first observations of this kind in the Eastern Mediterranean. The combination of these (and other Western European halocarbon) measurements with an atmospheric transport model, and Bayesian inversion techniques, allowed us to estimate regional-scale halocarbon emissions and for the first time provide reliable "top-town" emission estimates for the Eastern Mediterranean, a region of very diverse economic development and home to approximately 250 million people.

Due to the maritime and remote location of Finokalia, pollution from major metropolitan areas (the closest at a distance of 350-700km) tend to be better mixed into the background atmosphere at their arrival than at other continuous observation sites such as Monte Cimone (Italy) or Jungfraujoch (Switzerland). As expected this lead to generally smaller peak amplitudes for HFC-134a, HFC-125 and HFC-143a in Finokalia, compared to these sites. However, periodic peaks of HFC-152a were unexpectedly high, indicating one or several strong HFC-152a emission sources within the region directly influencing Finokalia. Higher peak mole fractions than at the Western European observation sites were observed for HCFC-22 and HCFC-142b, because of continued emissions from Article 5 regions such as Turkey, Egypt and the Middle East.

A range of sensitivity inversions showed that our regional-scale results are largely independent of the uncertainty assigned to and the absolute value of the a priori emissions and the design of the data-model-mismatch covariance matrix. Hence, for most compounds and emission regions the derived analytical a posteriori uncertainty was similar to the spread of the a posteriori emissions from all sensitivity inversions. In general, including off-diagonal elements in the uncertainty covariance matrices and, therefore, considering auto-correlation in the data-mismatch and a-priori uncertainty, led to lower a posteriori emission estimates (BASE and S-ML). Larger discrepancies between these sensitivity inversions were only seen for Central and Western Europe and HFC-134a emissions.

Our best estimate of a posteriori ("top-down") emissions and their uncertainties was derived as an average over the seven sensitivity inversions and considering their spread and individual

analytical uncertainty. For Article 5 countries in the Eastern Mediterranean (Turkey, Middle East, Egypt) a posteriori HCFC emissions were in the range assumed in our a priori, whereas they were smaller for the Non-Article 5 country Greece. In terms of HFC emissions in the Eastern Mediterranean, we estimated much smaller emissions than reported to the UNFCCC for all analysed compounds in Greece, whereas for Turkey our "top-down" estimates were similar to UNFCCC-reported values for HFC-125 and HFC-143a, but were much and slightly smaller for HFC-134a and HFC-152a, respectively. For the remaining regions in the Eastern Mediterranean no clear trend between "top-down" and our a priori estimates could be established, partly owing to the very insecure a priori estimates. For the Western and Central European areas of our inversion domain, our "top-down" estimates largely agree with other inverse modelling studies, although our results are within the lower range of previously reported emissions. Especially for HFC-134a and HFC-125 we obtained "top-down" estimates up to a factor of two smaller than reported UNFCCC values for the British Isles, France, Benelux and Germany.

In the context of lower-than-reported HFC-152a emissions from Turkey, the inversion algorithm was not able to perfectly simulate periodically measured, large HFC-152a pollution events at Finokalia. This could either be due to temporally varying emission sources, shortcomings in the atmospheric transport model or an unsuitable inversion setup. The latter two options can be ruled out since the transport simulation and inversion worked sufficiently well for other compounds. The first possibility was further analysed by using the temporal correlation between our observations and the simulated source sensitivity within individual grid cells during and around times when pollution events were observed. This allowed for the localisation of a possible emission region, located in the northwestern part of Turkey between the Agean coast and the city of Istanbul. The suspected temporal variability in the HFC-152a emissions rather points towards emissions from a HFC production plant than from product application and consumption.

Our measurements in Finokalia and the inversely estimated emissions show, that an additional observation site strongly increases the geographic extent and the quality of the inversion results, by reducing the a posteriori emission uncertainties in the Eastern Mediterranean in the range of 40-80% as compared to an inversion only using the Central European AGAGE observations. Including observations from Finokalia reduced estimated Greek HFC-134a emissions by a factor of four, while decreasing the uncertainty by the same factor. Additionally, the location of Finokalia allows the detection of Middle Eastern and North African emissions during specific

flow conditions, which is especially interesting due to the restrictions on the use of HCFCs for developing countries by the Montreal Protocol, which recently became effective. However, measurements during several years or a fixed monitoring station would be required to investigate trends in halocarbon emissions, for a continued "top-down" validation of South-Eastern European UNFCCC inventories or for the monitoring of the HCFC phase out in Eastern Mediterranean Article 5 countries.

# Acknowledgements

This research was funded by the Swiss National Science Foundation (project 200021-137638), the Swiss Federal Office for the Environment (FOEN), and the Swiss State Secretariat for Education and Research and Innovation (SERI). Additional funding was obtained from the EC FP7 project InGOS (Integrated Non-CO2 Greenhouse Gas Observing System; grant agreement number 284274) and trans-national access (TNA) from the EC FP7 ACTRIS Research Infra-

structure (grant agreement number 262254). We thank the Finokalia station staff for granting access to the site and supporting the setup and operation of our measurements. The International Foundation High Altitude Research Stations Jungfraujoch and Gornergrat (HFSJG) is acknowledged for the opportunity to perform observations at Jungfraujoch. The logistic at the "O. Vittori" station at Monte Cimone is supported by the National Research Council of Italy.

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

**Table 1: Setup for the base inversion (Base) and the sensitivity inversions (S-XX). Method refers to the uncertainty treatment explained in section 2.7. The sites are abbreviated as follows: Finokalia (FKL), Jungfraujoch (JFJ), Mace Head (MHD) and Monte Cimone (CMN).**

| Inversion | Method | Sites | Prior emissions uncertainty scaling factor | Prior emissions scaling factor |
|---|---|---|---|---|
| BASE | Global | FKL, JFJ, MHD, CMN | 1 | 1 |
| S-ML | Local | FKL, JFJ, MHD, CMN | 1 | 1 |
| S-MS | Stohl | FKL, JFJ, MHD, CMN | 1 | 1 |
| S-UH | Global | FKL, JFJ, MHD, CMN | 1.5 | 1 |
| S-UL | Global | FKL, JFJ, MHD, CMN | 0.5 | 1 |
| S-PH | Global | FKL, JFJ, MHD, CMN | 1 | 1.3 |
| S-PL | Global | FKL, JFJ, MHD, CMN | 1 | 0.7 |
| S-NFKL | Global | JFJ, MHD, CMN | 1 | 1 |
| S-OFKL | Global | FKL | 1 | 1 |

**Table 2: Inversion performance of the BASE inversion at Finokalia (FKL), Jungfraujoch (JFJ), Mace Head (MHD) and Monte Cimone (CMN). N is the number of observations used for the inversion. RMSE, $R^2$ and TSS denote the root mean square error, coefficient of determination and the Taylor skill score of the complete signal and $R^2_{abg}$ is the coefficient of determination of the signal above background.**

| | Site | N | RMSE (ppt) | | $R^2$ | | $R^2_{abg}$ | | TSS | |
|---|---|---|---|---|---|---|---|---|---|---|
| | | | apriori | apost | prior | post | prior | post | prior | post |
| HFC-134a | FKL | 1421 | 4.7 | 1.7 | 0.41 | 0.74 | 0.20 | 0.29 | 0.86 | 0.95 |
| | JFJ | 1946 | 4.5 | 3.6 | 0.33 | 0.50 | 0.25 | 0.34 | 0.82 | 0.71 |
| | MHD | 2005 | 3.3 | 2.9 | 0.61 | 0.74 | 0.61 | 0.73 | 0.93 | 0.75 |
| | CMN | 1801 | 5.8 | 5.1 | 0.39 | 0.54 | 0.25 | 0.28 | 0.62 | 0.74 |
| HFC-125 | FKL | 1147 | 1.4 | 0.8 | 0.31 | 0.59 | 0.12 | 0.16 | 0.81 | 0.88 |
| | JFJ | 1938 | 1.2 | 1.2 | 0.45 | 0.54 | 0.34 | 0.40 | 0.79 | 0.74 |
| | MHD | 1975 | 1.0 | 0.8 | 0.63 | 0.74 | 0.62 | 0.71 | 0.88 | 0.89 |
| | CMN | 1840 | 1.8 | 1.6 | 0.42 | 0.53 | 0.29 | 0.32 | 0.62 | 0.76 |
| HFC-152a | FKL | 1428 | 4.0 | 1.2 | 0.00 | 0.43 | 0.00 | 0.11 | 0.39 | 0.62 |
| | JFJ | 1960 | 1.4 | 1.3 | 0.36 | 0.49 | 0.21 | 0.31 | 0.59 | 0.65 |
| | MHD | 2011 | 0.7 | 0.5 | 0.54 | 0.72 | 0.26 | 0.38 | 0.89 | 0.90 |
| | CMN | 1864 | 1.5 | 1.3 | 0.33 | 0.55 | 0.19 | 0.29 | 0.54 | 0.74 |
| HFC-143a | FKL | 1252 | 2.3 | 1.6 | 0.06 | 0.53 | 0.01 | 0.03 | 0.26 | 0.67 |
| | JFJ | 1973 | 1.2 | 1.1 | 0.43 | 0.48 | 0.36 | 0.38 | 0.83 | 0.70 |
| | MHD | 2052 | 1.1 | 0.9 | 0.63 | 0.72 | 0.65 | 0.71 | 0.79 | 0.85 |
| | CMN | 1814 | 1.5 | 1.4 | 0.41 | 0.48 | 0.31 | 0.31 | 0.74 | 0.72 |
| HCFC-22 | FKL | 1426 | 3.7 | 2.7 | 0.15 | 0.42 | 0.05 | 0.15 | 0.52 | 0.62 |
| | JFJ | 1953 | 2.8 | 2.1 | 0.31 | 0.50 | 0.14 | 0.23 | 0.77 | 0.73 |
| | MHD | 1994 | 1.8 | 1.3 | 0.41 | 0.65 | 0.26 | 0.36 | 0.84 | 0.89 |
| | CMN | 1728 | 3.0 | 2.3 | 0.35 | 0.50 | 0.15 | 0.19 | 0.76 | 0.76 |
| HCFC- | FKL | 1065 | 0.6 | 0.5 | 0.52 | 0.64 | 0.00 | 0.02 | 0.79 | 0.87 |
| | JFJ | 1960 | 0.4 | 0.3 | 0.24 | 0.39 | 0.12 | 0.15 | 0.36 | 0.62 |
| | MHD | 2042 | 0.2 | 0.1 | 0.42 | 0.66 | 0.36 | 0.52 | 0.64 | 0.84 |

| CMN | 1802 | 0.4 | 0.3 | 0.48 | 0.56 | 0.21 | 0.19 | 0.57 | 0.82 |

**Table 3: Regional emissions as estimated in the a priori inventory and by the atmospheric inversion. All values are given in Gg yr$^{-1}$. A posteriori estimates are shown as the mean values, derived from the BASE inversion and the sensitivity inversions S-ML, S-MS, S-UH, S-UL, S-PH, S-PL. The uncertainty range gives the maximum range provided by the respective mean values of all inversions plus the mean of the analytic uncertainty ($p < 0.05$) estimated by each individual inversion. Smaller and distant countries were aggregated to larger regions: Turkey (Turkey, Cyprus), Balkans (Serbia, Montenegro, Kosovo, Albania, Bosnia and Herzegovina, Croatia, Slovenia, FYROM), Eastern (Ukraine, Romania, Moldova, Bulgaria), Middle East (Jordan, Lebanon, Syria, Palestine, Israel), Maghreb (Morocco, Algeria, Tunisia, Libya), Central E (Poland, Slovakia, Czech-Republic, Hungary), Central W (Switzerland, Liechtenstein, Germany, Austria, Denmark), Western (France, Luxembourg, Netherlands, Belgium), Iberian Peninsula (Spain, Portugal), British Isles (Ireland, United Kingdom).**

| | HFC-134a ($CH_2FCF_3$) | | HFC-125 ($C_2HF_5$) | | HFC-152a ($C_2H_4F_2$) | |
|---|---|---|---|---|---|---|
| | Prior | Post | Prior | Post | Prior | Post |
| Greece | 1.32±0.53 | 0.40 (0.26-0.63) | 0.60±0.24 | 0.24 (0.17-0.32) | 1.22±0.49 | 0.23 (0.13-0.37) |
| Turkey | 2.85±1.16 | 1.42 (0.86-1.97) | 0.12±0.05 | 0.11 (0.06-0.16) | 1.10±0.78 | 0.64 (0.37-1.05) |
| Balkans | 0.65±1.03 | 0.70 (0.29-1.10) | 0.12±0.19 | 0.18 (0.05-0.31) | 0.12±0.20 | 0.19 (0.11-0.35) |
| Eastern | 1.15±0.92 | 0.84 (0.29-1.39) | 0.34±0.28 | 0.31 (0.10-0.53) | 0.18±0.18 | 0.14 (0.01-0.27) |
| Middle East | 0.65±1.26 | 0.22 (-0.26-0.70) | 0.14±0.28 | 0.15 (-0.10-0.41) | 0.23±0.44 | 0.19 (-0.02-0.40) |
| Egypt | 1.14±2.28 | 0.90 (0.28-1.51) | 0.28±0.56 | 0.20 (-0.05-0.45) | 0.46±0.92 | 0.08 (-0.14-0.31) |
| Maghreb | 1.18±2.34 | 0.90 (-0.02-1.82) | 0.30±0.59 | 0.39 (0.03-0.75) | 0.47±0.93 | 0.16 (-0.02-0.33) |
| Central E | 2.64±1.06 | 1.53 (1.03-2.03) | 1.10±0.44 | 0.74 (0.53-0.96) | 0.41±0.16 | 0.28 (0.18-0.37) |
| Central W | 5.67±2.28 | 2.33 (1.73-3.18) | 0.95±0.38 | 0.68 (0.50-0.90) | 0.34±0.18 | 0.25 (0.17-0.37) |
| Western | 6.07±2.42 | 3.10 (2.38-3.84) | 1.92±0.77 | 1.40 (1.19-1.61) | 0.44±0.17 | 0.30 (0.23-0.37) |
| Italy | 1.96±0.79 | 1.85 (1.58-2.13) | 1.06±0.42 | 1.05 (0.91-1.19) | 0.01±0.00 | 0.01 (0.01-0.02) |
| Iberian Pen. | 4.06±1.63 | 1.82 (1.16-2.58) | 2.02±0.81 | 1.50 (1.19-1.82) | 0.32±0.13 | 0.20 (0.11-0.29) |
| British Isles | 5.22±2.09 | 2.63 (2.12-3.54) | 1.49±0.60 | 1.09 (0.97-1.22) | 0.21±0.08 | 0.10 (0.06-0.14) |
| Domain Total | 34.56±5.97 | 18.64 (16.70-20.57) | 10.45±1.74 | 8.07 (7.30-8.83) | 5.49±1.71 | 2.77 (2.27-3.27) |
| | HFC-143a ($C_2H_3F_3$) | | HCFC-22 ($CHClF_2$) | | HCFC-142b ($C_2H_3ClF_2$) | |
| | Prior | Post | Prior | Post | Prior | Post |
| Greece | 0.17±0.07 | 0.11 (0.06-0.15) | 0.20±0.16 | 0.13 (0.04-0.23) | 0.016±0.013 | 0.015 (0.003-0.026) |
| Turkey | 0.05±0.02 | 0.04 (0.02-0.06) | 1.38±2.78 | 0.83 (0.02-1.65) | 0.112±0.157 | 0.140 (0.025-0.256) |
| Balkans | 0.08±0.13 | 0.12 (0.03-0.21) | 0.45±0.36 | 0.28 (0.07-0.50) | 0.036±0.029 | 0.041 (0.017-0.064) |
| Eastern | 0.09±0.07 | 0.09 (0.03-0.16) | 1.51±1.22 | 0.62 (-0.01-1.25) | 0.122±0.099 | 0.071 (-0.004-0.146) |
| Middle East | 0.10±0.19 | 0.09 (-0.07-0.26) | 0.78±1.52 | 0.97 (0.26-1.82) | 0.063±0.086 | 0.059 (-0.025-0.143) |
| Egypt | 0.20±0.39 | 0.24 (0.04-0.44) | 1.55±3.09 | 2.08 (1.27-2.89) | 0.125±0.175 | 0.056 (-0.059-0.170) |
| Maghreb | 0.21±0.41 | 0.41 (0.15-0.67) | 1.57±3.13 | 0.54 (-0.01-1.08) | 0.127±0.177 | 0.052 (-0.013-0.116) |
| Central (E) | 0.92±0.37 | 0.60 (0.44-0.77) | 1.18±0.94 | 0.35 (0.01-0.70) | 0.095±0.076 | 0.051 (0.008-0.094) |
| Central (W) | 0.65±0.26 | 0.52 (0.39-0.68) | 1.87±1.50 | 0.60 (0.24-0.98) | 0.151±0.121 | 0.126 (0.085-0.167) |
| Western | 1.49±0.60 | 1.20 (1.06-1.35) | 1.67±1.33 | 0.75 (0.45-1.04) | 0.135±0.108 | 0.186 (0.151-0.221) |
| Italy | 0.92±0.37 | 0.71 (0.61-0.82) | 1.08±0.87 | 0.69 (0.47-0.91) | 0.088±0.070 | 0.095 (0.065-0.124) |
| Iberian Pen. | 1.00±0.40 | 0.93 (0.73-1.13) | 1.03±0.83 | 0.38 (0.02-0.73) | 0.083±0.067 | 0.050 (0.006-0.094) |
| British Isles | 0.76±0.31 | 0.70 (0.61-0.79) | 1.24±1.00 | 0.67 (0.49-0.84) | 0.101±0.080 | 0.070 (0.050-0.089) |
| Domain Total | 6.65±1.16 | 5.77 (5.25-6.30) | 15.51±6.20 | 8.89 (7.12-10.66) | 1.253±0.391 | 1.009 (0.782-1.237) |

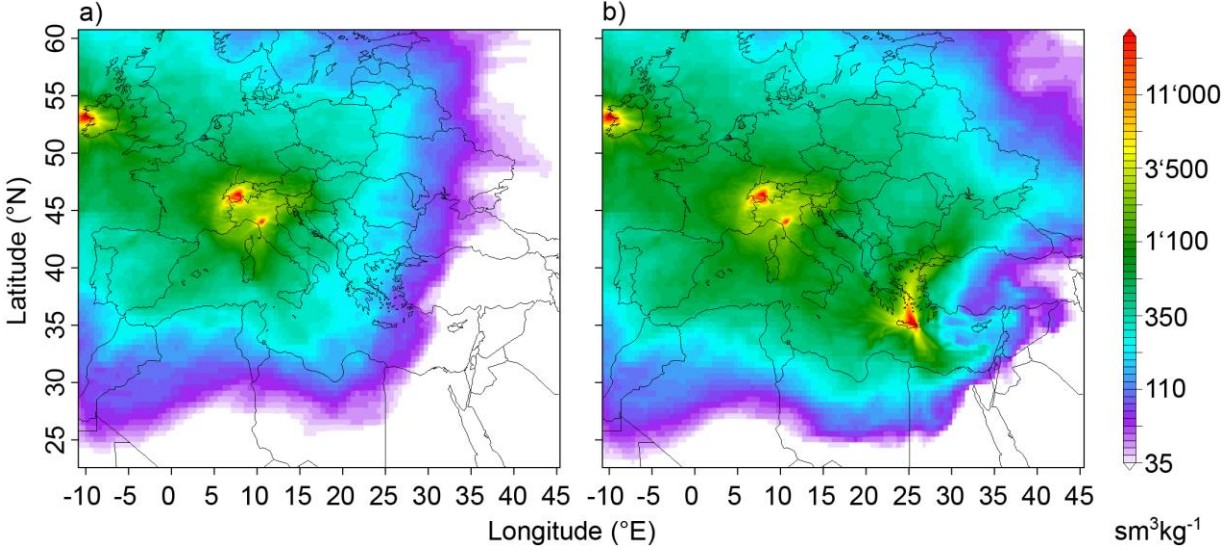

**Figure 1: Average FLEXPART derived source sensitivities for the inversions period and domain for a) the measurements at the AGAGE stations Mace Head (MHD), Jungfraujoch (JFJ) and Monte Cimone (CMN) and b) the additional measurements at Finokalia (FKL).**

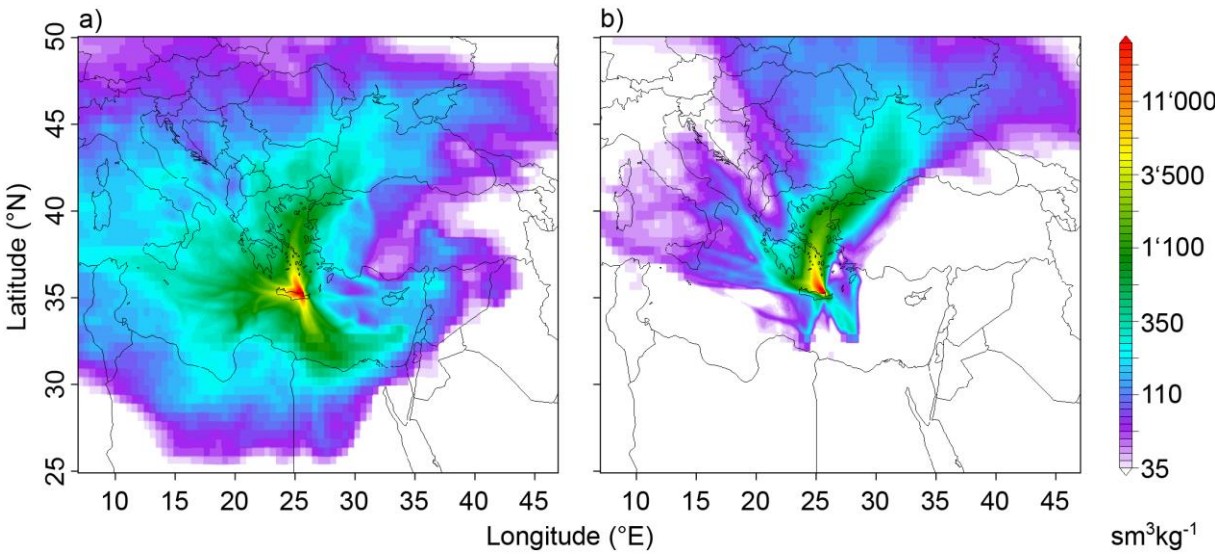

**Figure 2: Average FLEXPART derived source sensitivities for Finokalia and two characteristic flow regimes during the measurement campaign: a) shows the variable flow during winter and spring and b) northeasterly flow during the summer months.**

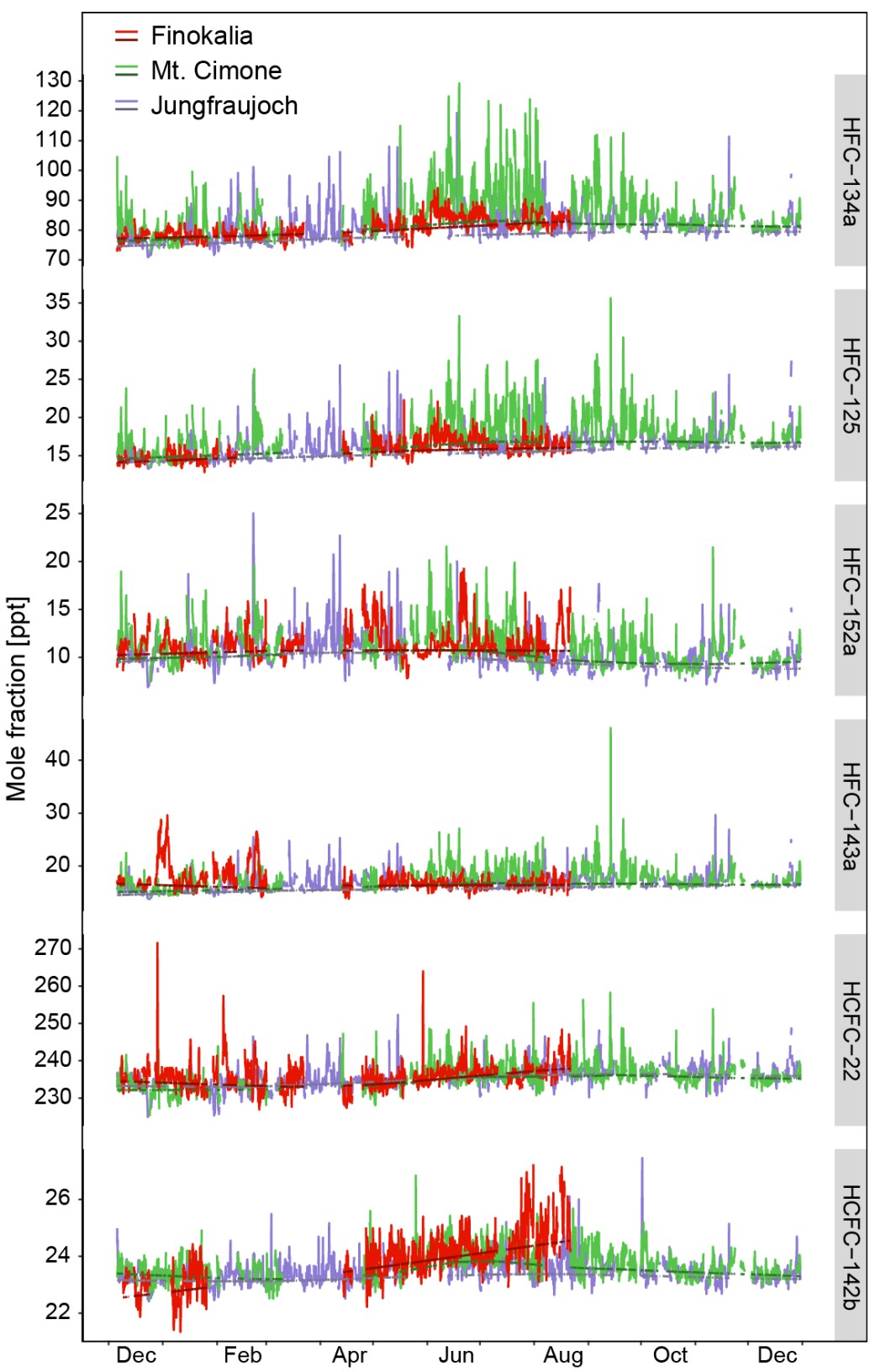

**Figure 3: Halocarbon observations in 2013, during the time of the measurement campaign in Finokalia (red) and simultaneous measurements at Jungfraujoch (purple) and Monte Cimone (green). The corresponding background estimated with REBS is shown in the darker shade of the respective color (ppt refers to SI unit pmol mol⁻¹).**

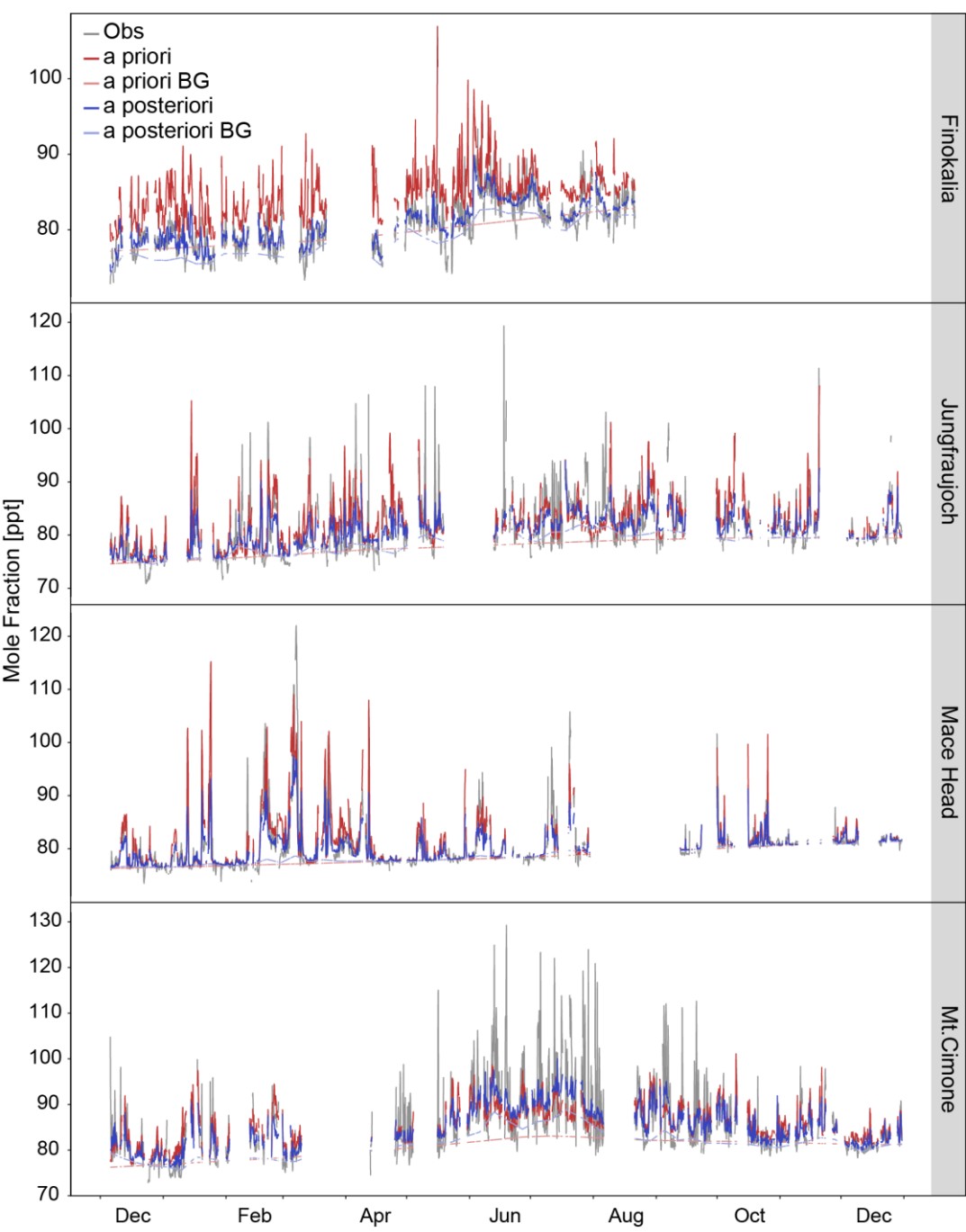

**Figure 4: HFC-134a time series of the base inversion for 2013, showing the observed mole fractions at the respective sites (grey) and the simulated values (a priori: red; a posteriori: blue) and their baseline conditions (a priori: light red; a posteriori: light blue).**

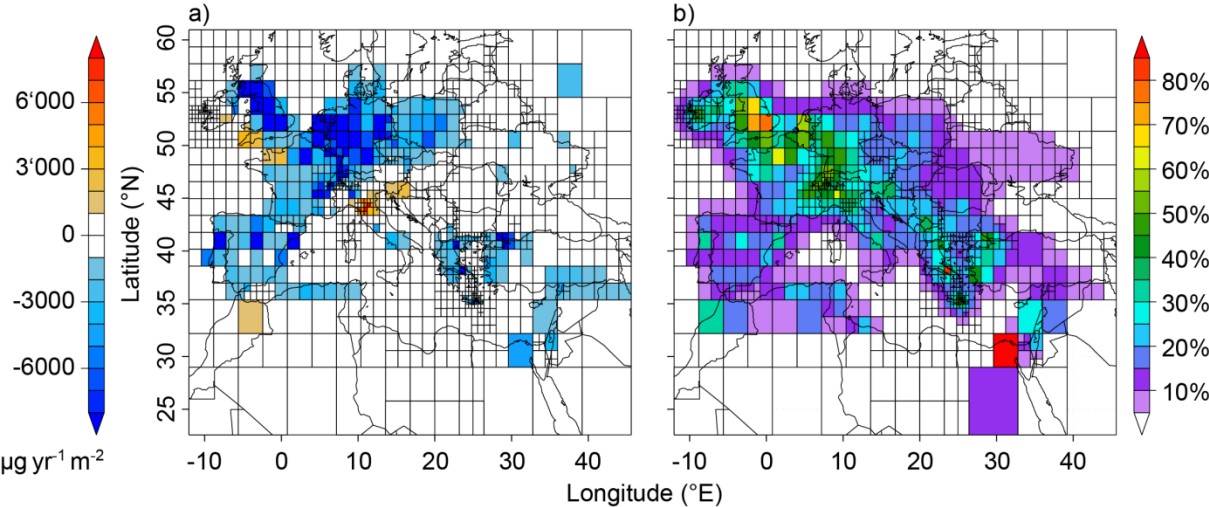

**Figure 5: (a) Emissions difference (posterior – prior) of the BASE inversion of HFC-134a. (b) Relative reduction of the a posteriori uncertainty compared to the a priori uncertainties of HFC-134a.**

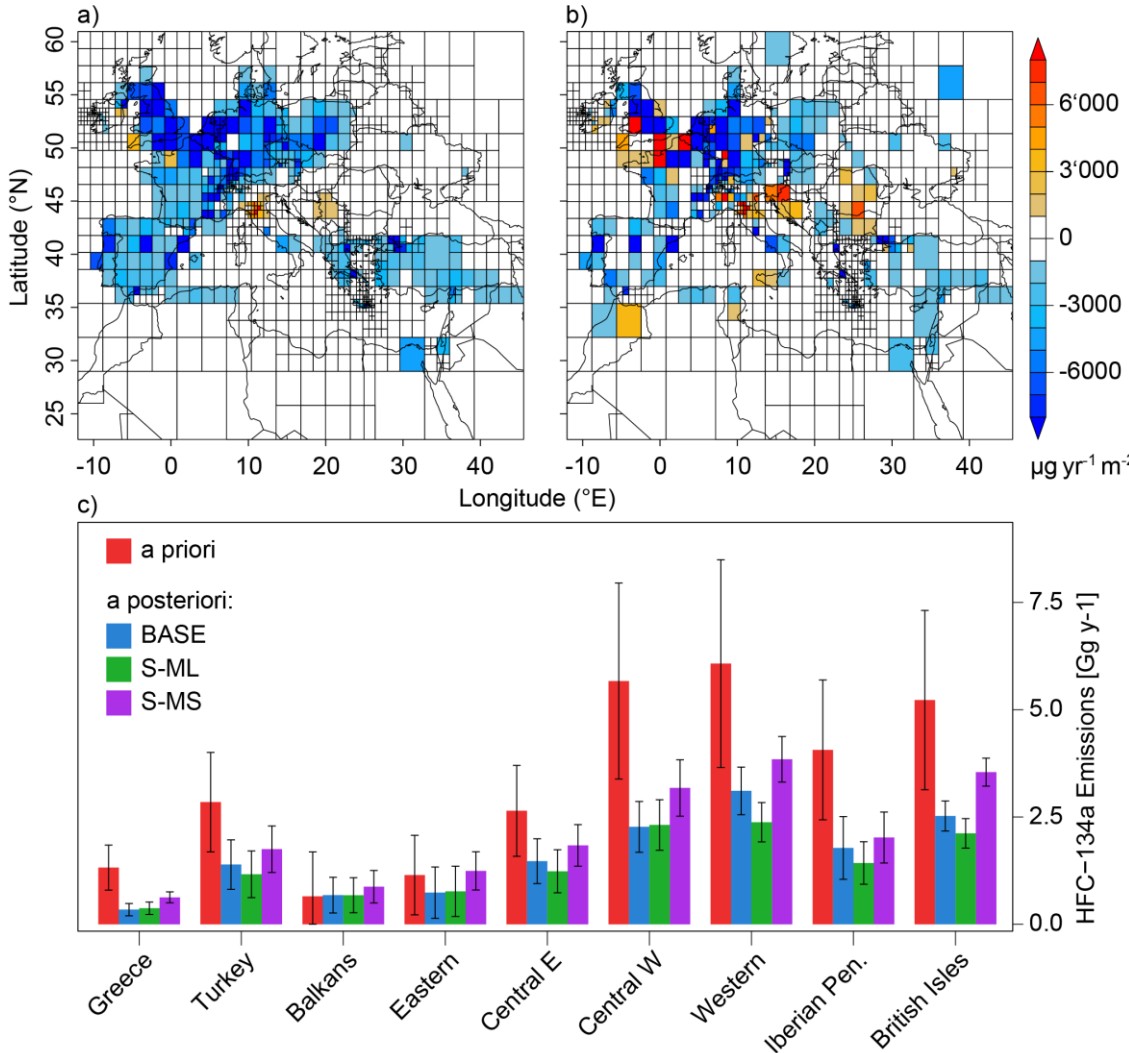

**Figure 6: Difference of the a posteriori and a priori emissions for (a) the S-ML and (b) the S-MS inversions of HFC-134a. (c) regional emission estimates: a priori emissions (red) and a posteriori emissions (BASE = green, S-ML = blue, S-MS = purple). The uncertainties given are two standard deviations of the analytic uncertainty assigned to the a priori emissions and derived by the inversion as a posteriori uncertainties.**

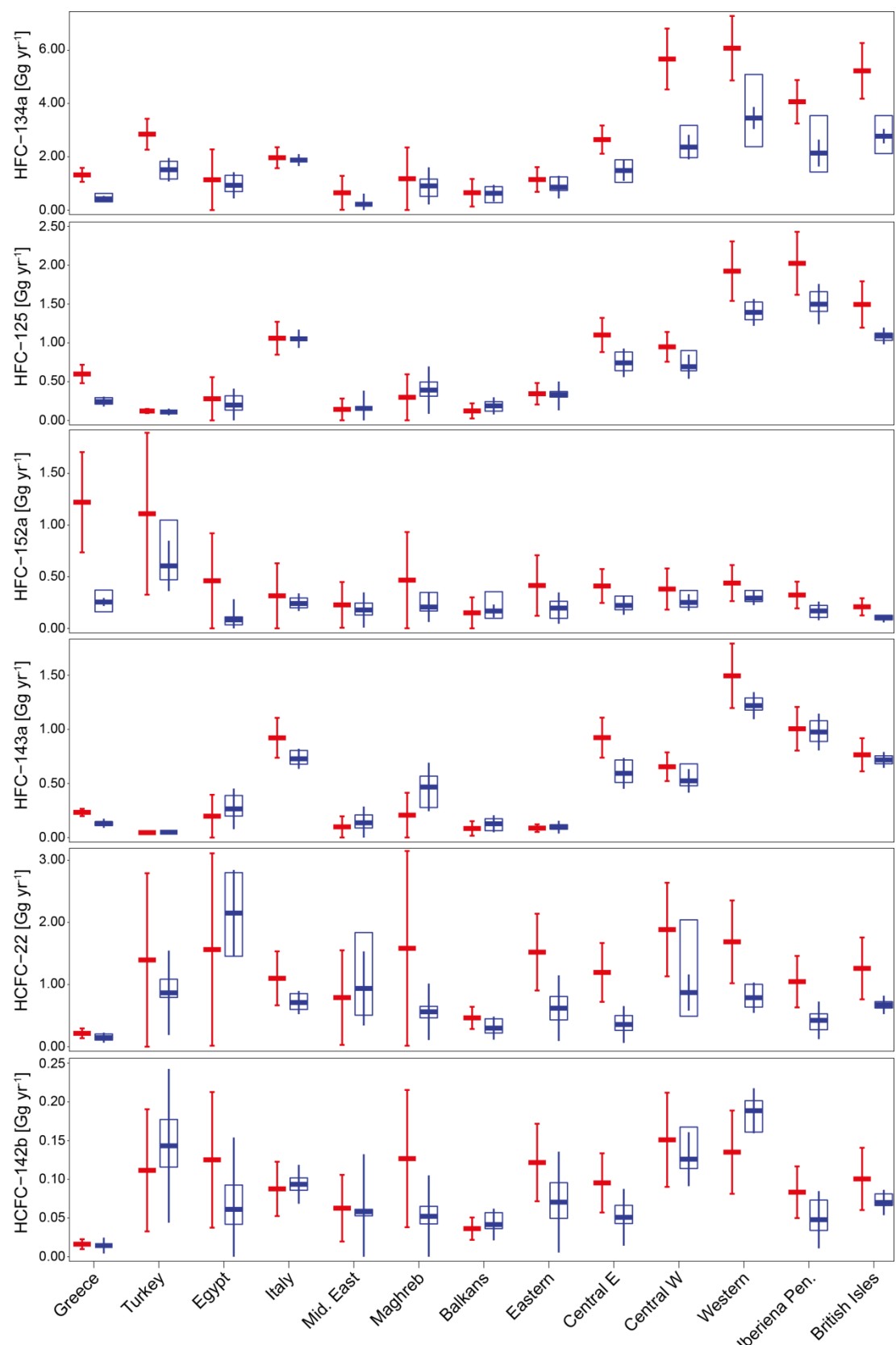

**Figure 7: Annual emissions of 2013 for the aggregated regions. A priori emissions are shown in red, with uncertainty giving the 95% confidence range. For the a posteriori estimates boxes show the range of all sensitivity inversions, whereas the thick horizontal line gives the mean of all sensitivity inversions. In addition, the blue error bars give the analytic uncertainty (95% confidence level) averaged over all uncertainty inversions.**

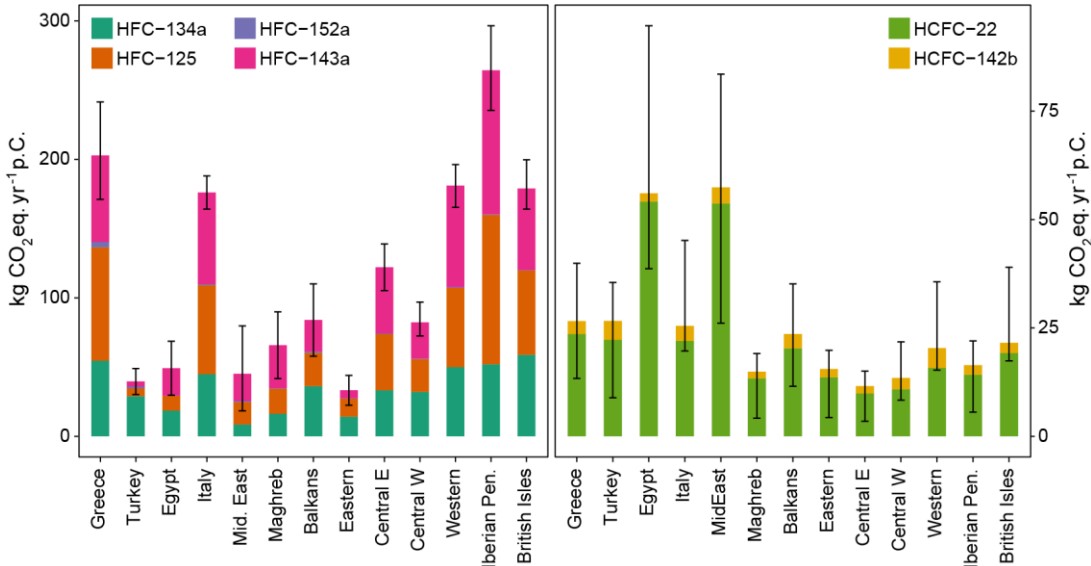

**Figure 8: Annual per capita (p.C.) emissions in CO₂ equivalents, derived from the base inversion and all sensitivity inversions (best estimate). The results have been computed using the 100-yr GWP (GWP$_{100}$) values of [*Harris and Wuebbles*, 2014]. The bars show the average mean of all inversions, whereas the error bars show our uncertainty estimate including analytical and structural uncertainty.**

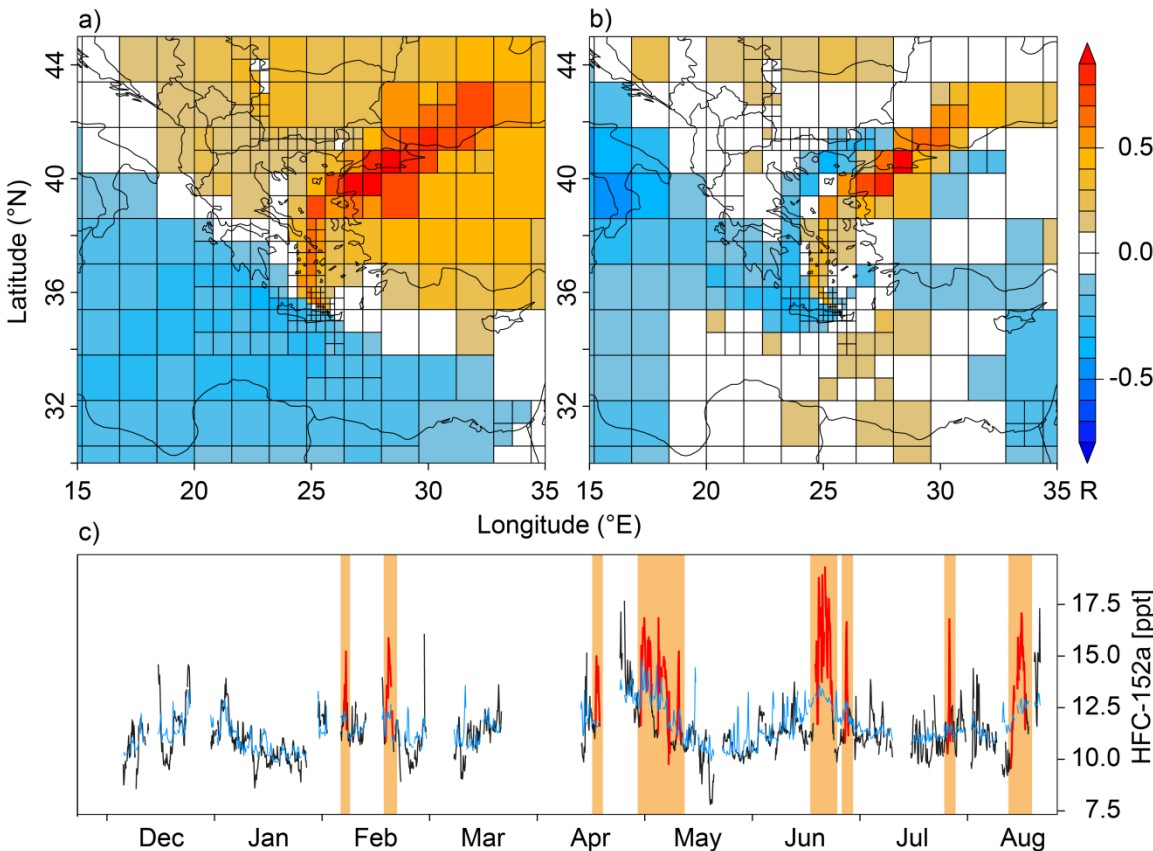

**Figure 9: Spatial distribution of the Pearson Correlation Coefficient (R) for (a) the entire time series of HFC-152a observations at Finokalia and the per-cell source sensitivity and for (b) the period of the pollution peaks, which are highlighted in red in (c) the observed (black) and simulated a posteriori (BASE inversion) (blue) mole fractions of HFC-152a.**

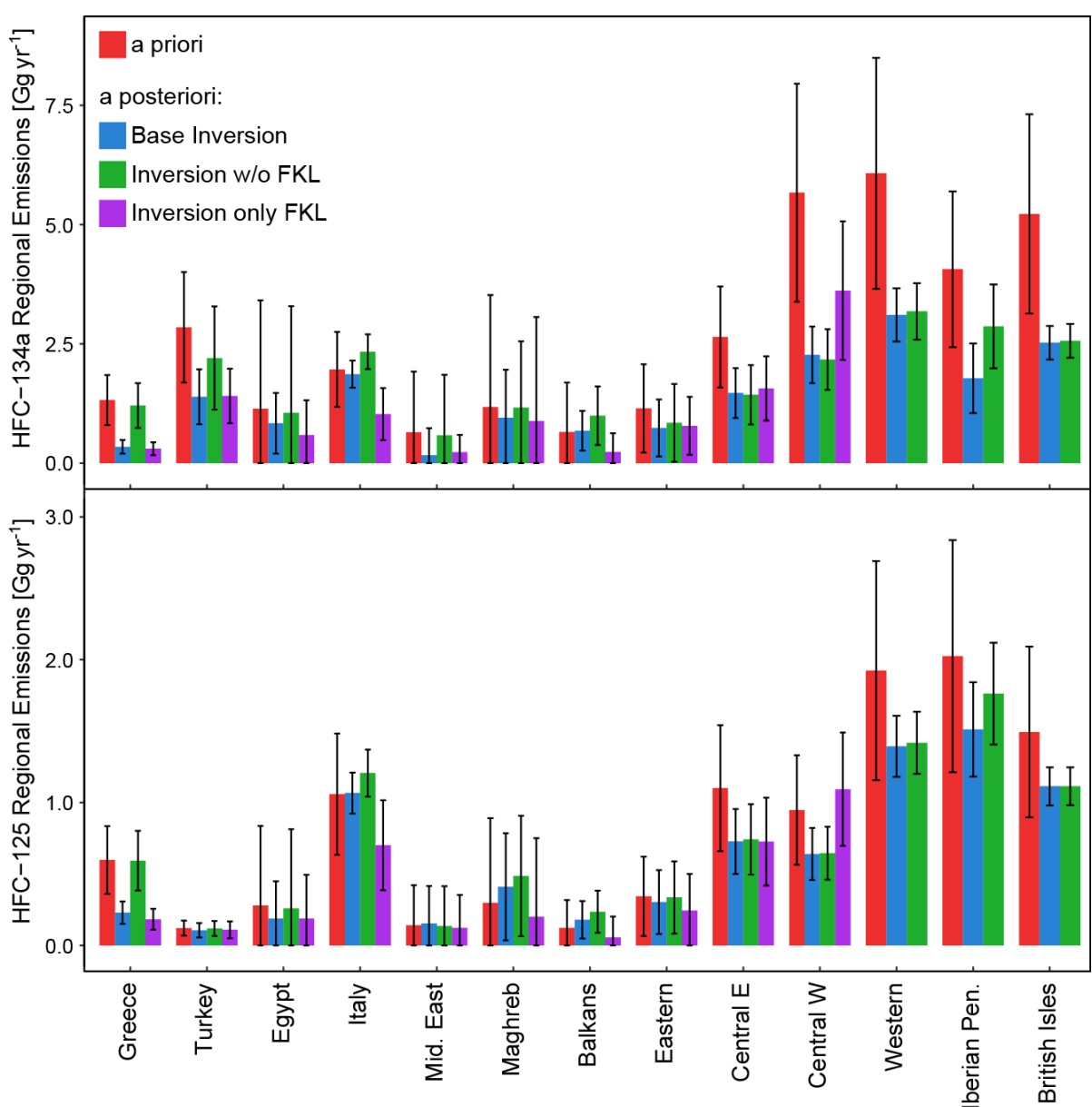

**Figure 10: Regional annual emission estimates of 2013. The apriori of our base inversion is shown in red. A posteriori results are shown for the BASE inversion (blue), the inversion excluding Finokalia (S-NFKL, green) and the inversion using only observations from Finokalia (S-OFKL, purple). Error bars represent the 95% confidence levels. Note that for the inversion based on Finokalia observations alone (S-OFKL) the inversion domain was cropped in the West and no a posteriori emission for the western part of the domain were estimated.**

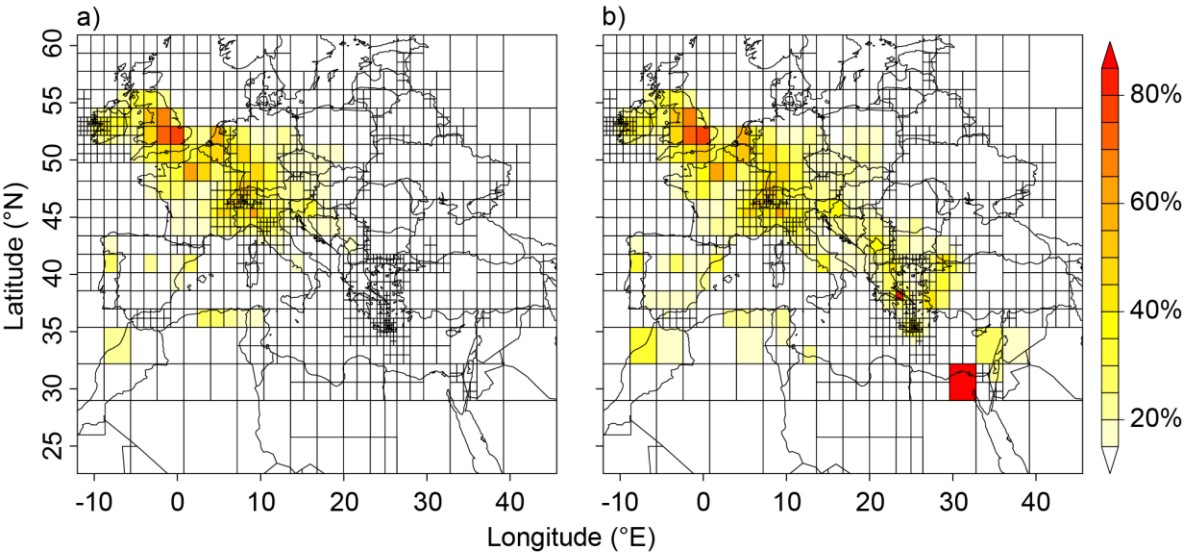

**Figure 11: HFC-134a uncertainty reduction (%) achieved by (a) the inversion excluding observations from Finokalia (S-NFKL) and (b) the BASE inversion using observations from all four sites including Finokalia.**

**SI-Table 1: Basic statistics for the 3-hourly aggregates of the observations taken at all sites during the campaign period (Dec. 2012 – Aug.2013). Observation sites are: Finokalia (FKL), Jungfraujoch (JFJ), Mace Head (MHD), Monte Cimone (CMN). Shown are the number of observations (N), the mean, minimum (Min), maximum (Max) and standard deviation (SD) for the observations and the baseline values, estimated with REBS. The mean measurement uncertainty ($\sigma_O$) was determined from the standard deviation of reference gas measurements and the baseline uncertainty ($\sigma_b$) was derived as one constant value by the REBS method.**

| | Site | N | Observations | | | | | Background (REBS) | | | |
|---|---|---|---|---|---|---|---|---|---|---|---|
| | | | Mean [ppt] | Min [ppt] | Max [ppt] | SD [ppt] | $\sigma_o$ [ppt] | Mean [ppt] | Min [ppt] | Max [ppt] | $\sigma_b$ [ppt] |
| HFC-134a | FKL | 1467 | 80.8 | 72.8 | 94.2 | 3.4 | 0.8 | 79.7 | 77.2 | 83.0 | 1.9 |
| | JFJ | 1383 | 80.7 | 70.9 | 119.3 | 5.3 | 0.2 | 77.0 | 74.6 | 79.1 | 1.3 |
| | MHD | 1533 | 80.3 | 73.5 | 122.0 | 5.6 | 0.2 | 77.4 | 76.3 | 78.9 | 0.7 |
| | CMN | 1040 | 86.1 | 72.8 | 129.3 | 8.7 | 0.3 | 80.1 | 76.2 | 83.1 | 1.7 |
| HFC-125 | FKL | 1193 | 15.9 | 12.8 | 22.3 | 1.3 | 0.4 | 15.3 | 14.1 | 16.1 | 0.6 |
| | JFJ | 1373 | 16.1 | 13.3 | 26.9 | 1.7 | 0.1 | 14.9 | 14.1 | 15.7 | 0.3 |
| | MHD | 1514 | 15.8 | 13.9 | 28.1 | 1.7 | 0.1 | 14.9 | 14.5 | 15.4 | 0.2 |
| | CMN | 1078 | 17.6 | 13.4 | 33.3 | 2.6 | 0.1 | 15.8 | 14.6 | 16.8 | 0.6 |
| HFC-152a | FKL | 1428 | 11.5 | 7.8 | 19.3 | 1.6 | 0.2 | 10.6 | 10.3 | 10.7 | 0.8 |
| | JFJ | 1395 | 10.8 | 6.9 | 25.0 | 1.7 | 0.1 | 10.0 | 9.4 | 10.5 | 0.8 |
| | MHD | 1527 | 10.9 | 8.4 | 15.0 | 0.9 | 0.1 | 10.6 | 9.7 | 10.8 | 0.4 |
| | CMN | 1096 | 11.7 | 7.3 | 21.6 | 1.9 | 0.1 | 10.3 | 9.7 | 10.7 | 0.7 |
| HFC-143a | FKL | 1252 | 17.4 | 13.2 | 29.6 | 2.2 | 1.2 | 16.3 | 15.8 | 16.7 | 0.9 |
| | JFJ | 1411 | 16.7 | 13.7 | 25.6 | 1.6 | 0.1 | 15.4 | 14.5 | 16.0 | 0.3 |
| | MHD | 1540 | 16.6 | 14.8 | 27.6 | 1.8 | 0.1 | 15.5 | 15.3 | 15.9 | 0.2 |
| | CMN | 1055 | 17.5 | 14.2 | 27.1 | 1.9 | 0.1 | 16.0 | 15.1 | 16.8 | 0.5 |
| HCFC-22 | FKL | 1438 | 235.8 | 226.9 | 271.6 | 3.5 | 1.8 | 234.7 | 233.0 | 237.9 | 2.3 |
| | JFJ | 1389 | 234.9 | 224.9 | 252.4 | 2.9 | 0.6 | 234.2 | 233.0 | 236.3 | 2.2 |
| | MHD | 1523 | 235.8 | 230.3 | 259.8 | 1.8 | 0.6 | 235.2 | 235.0 | 236.0 | 1.1 |
| | CMN | 980 | 235.1 | 225.2 | 255.5 | 3.3 | 0.7 | 234.0 | 232.2 | 235.8 | 1.9 |
| HCFC-142b | FKL | 1075 | 23.9 | 21.3 | 27.2 | 0.9 | 0.6 | 23.7 | 22.6 | 24.5 | 0.5 |
| | JFJ | 1392 | 23.4 | 22.4 | 26.1 | 0.4 | 0.1 | 23.2 | 23.1 | 23.4 | 0.2 |
| | MHD | 1533 | 23.3 | 22.7 | 24.7 | 0.2 | 0.1 | 23.2 | 23.1 | 23.3 | 0.1 |
| | CMN | 1046 | 23.8 | 22.5 | 26.8 | 0.5 | 0.1 | 23.5 | 23.2 | 23.8 | 0.3 |

**SI-Table 2: Inversion performance of the base inversion and the sensitivity inversions S-ML and S-MS for HFC-134a**
**at Finokalia (FKL), Jungfraujoch (JFJ), Mace Head (MHD) and Monte Cimone (CMN). N is the number of observa-**
**tions used for the inversion. RMSE is the root mean square error in ppt (parts per billion $10^{-12}$). $R^2$ denotes the coeffi-**
**cient of determination of the complete signals and $R^2_{abg}$ is the coefficient of determination of the signals above back-**
**ground. TSS shows the Taylor Skill Score of the entire signal.**

| | Site | N | RMSE | | $R^2$ | | $R^2_{abg}$ | | TSS | |
|---|---|---|---|---|---|---|---|---|---|---|
| | | | apriori | apost | prior | post | prior | post | prior | post |
| Base | FKL | 1421 | 4.7 | 1.7 | 0.41 | 0.74 | 0.20 | 0.29 | 0.86 | 0.95 |
| | JFJ | 1946 | 4.5 | 3.6 | 0.33 | 0.50 | 0.25 | 0.34 | 0.82 | 0.71 |
| | MHD | 2005 | 3.3 | 2.9 | 0.61 | 0.74 | 0.61 | 0.73 | 0.93 | 0.75 |
| | CMN | 1801 | 5.8 | 5.1 | 0.39 | 0.54 | 0.25 | 0.28 | 0.62 | 0.74 |
| S-ML | FKL | 1421 | 4.7 | 1.7 | 0.41 | 0.75 | 0.20 | 0.28 | 0.86 | 0.95 |
| | JFJ | 1946 | 4.5 | 3.6 | 0.33 | 0.50 | 0.25 | 0.32 | 0.82 | 0.68 |
| | MHD | 2005 | 3.3 | 3.3 | 0.61 | 0.68 | 0.61 | 0.68 | 0.93 | 0.66 |
| | CMN | 1801 | 5.8 | 5.1 | 0.39 | 0.55 | 0.25 | 0.28 | 0.62 | 0.73 |
| S-MS | FKL | 1421 | 4.7 | 1.7 | 0.41 | 0.75 | 0.20 | 0.36 | 0.86 | 0.97 |
| | JFJ | 1946 | 4.5 | 3.4 | 0.33 | 0.53 | 0.25 | 0.4 | 0.82 | 0.80 |
| | MHD | 2005 | 3.3 | 2.5 | 0.61 | 0.76 | 0.61 | 0.75 | 0.93 | 0.90 |
| | CMN | 1801 | 5.8 | 5.0 | 0.39 | 0.55 | 0.25 | 0.31 | 0.62 | 0.78 |

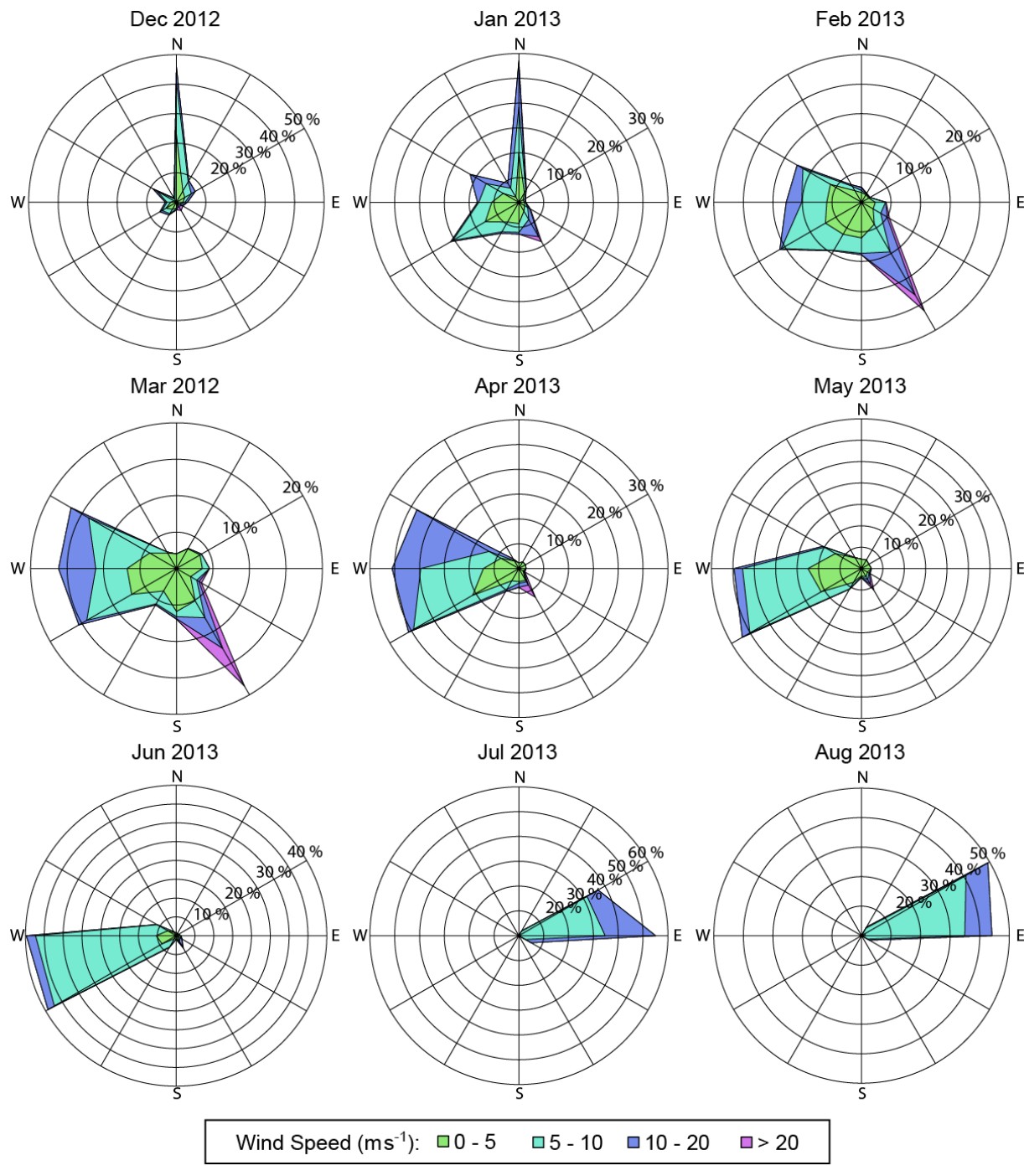

**SI-Figure 1: Observed monthly wind roses at Finokalia for the period January to August 2013 showing the directional**
**frequencies colour-coded by wind speed based on 5 minute temporal resolution. Wind data were provided by the Uni-**
**versity of Crete.**

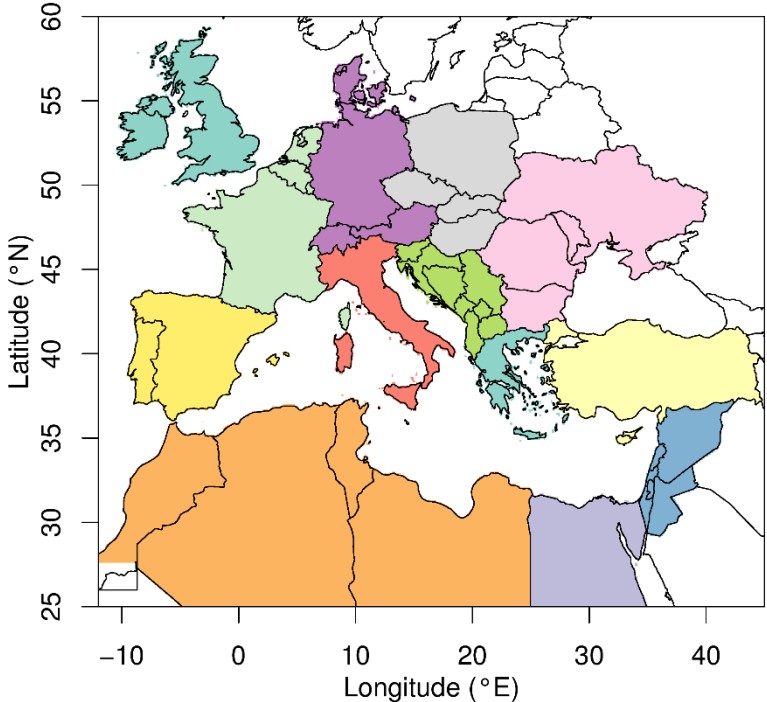

**SI-Figure 2: Illustration of region definition used in the discussion of emission estimates: Greece (light turquoise), Tur-**
**key (Turkey, Cyprus; pale yellow), Balkans (Serbia, Montenegro, Kosovo, Albania, Bosnia and Herzegovina, Croatia,**
**Slovenia, Former Yugoslav Republic of Macedonia (FYROM); light green), Eastern (Ukraine, Romania, Moldova, Bul-**
**garia; pale pink), Middle East (Jordan, Lebanon, Syria, Palestine, Israel; blue), Egypt (pale purple), Maghreb (Mo-**
**rocco, Algeria, Tunisia, Libya; orange), Central E (Poland, Slovakia, Czech-Republic, Hungary; grey), Central W (Swit-**
**zerland, Liechtenstein, Germany, Austria, Denmark; purple), Western (France, Luxembourg, Netherlands, Belgium;**
**pale green), Italy (red), Iberian Peninsula (Spain, Portugal; yellow), British Isles (Ireland, United Kingdom; light tur-**
**quoise).**

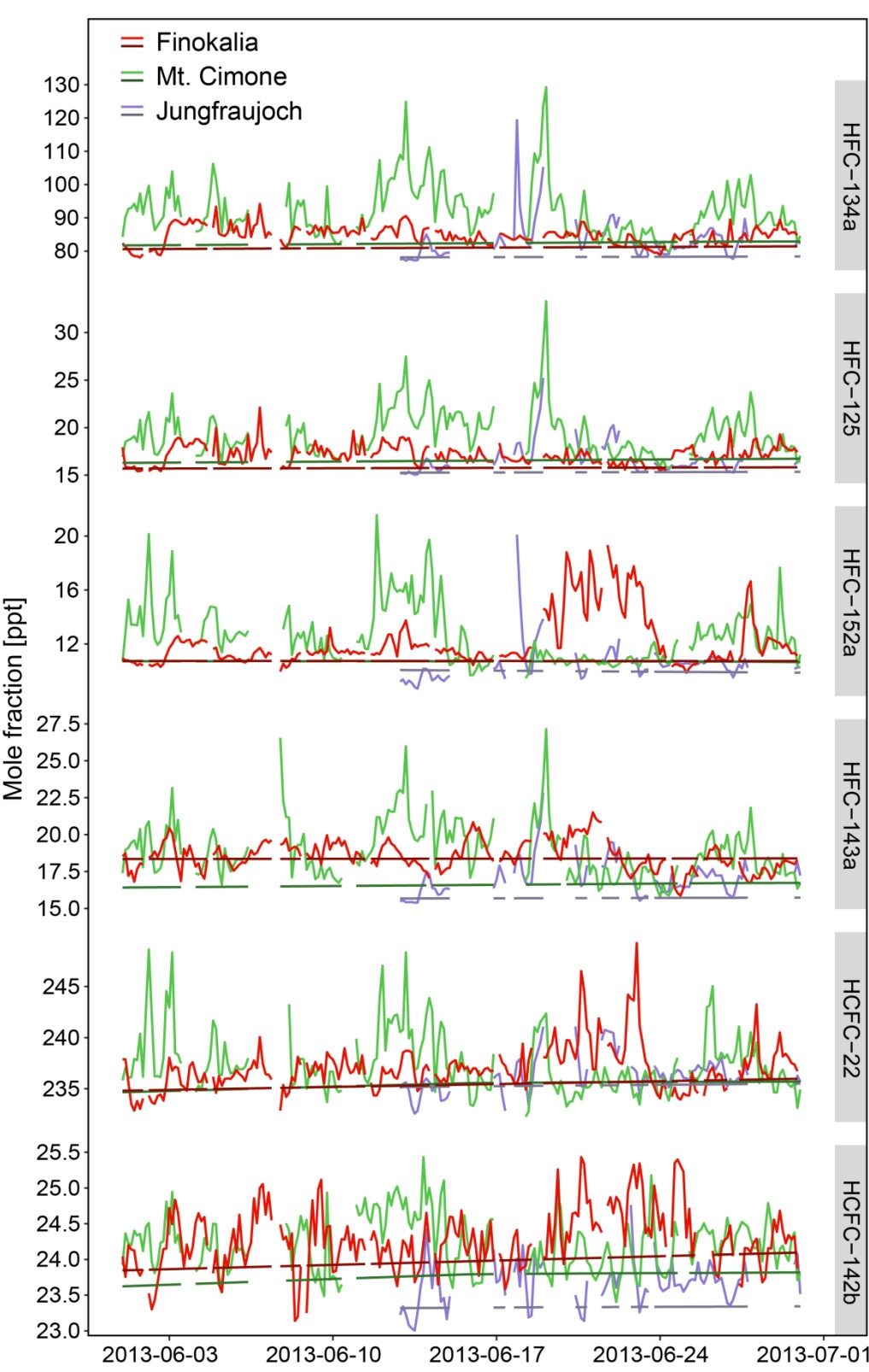

**SI-Figure 3: Halocarbon observations in June 2013 at Finokalia (red) and simultaneous measurements at Jungfraujoch**
**(purple) and Monte Cimone (green). The corresponding background estimated with REBS is shown in the darker shade**
**of the respective color.**

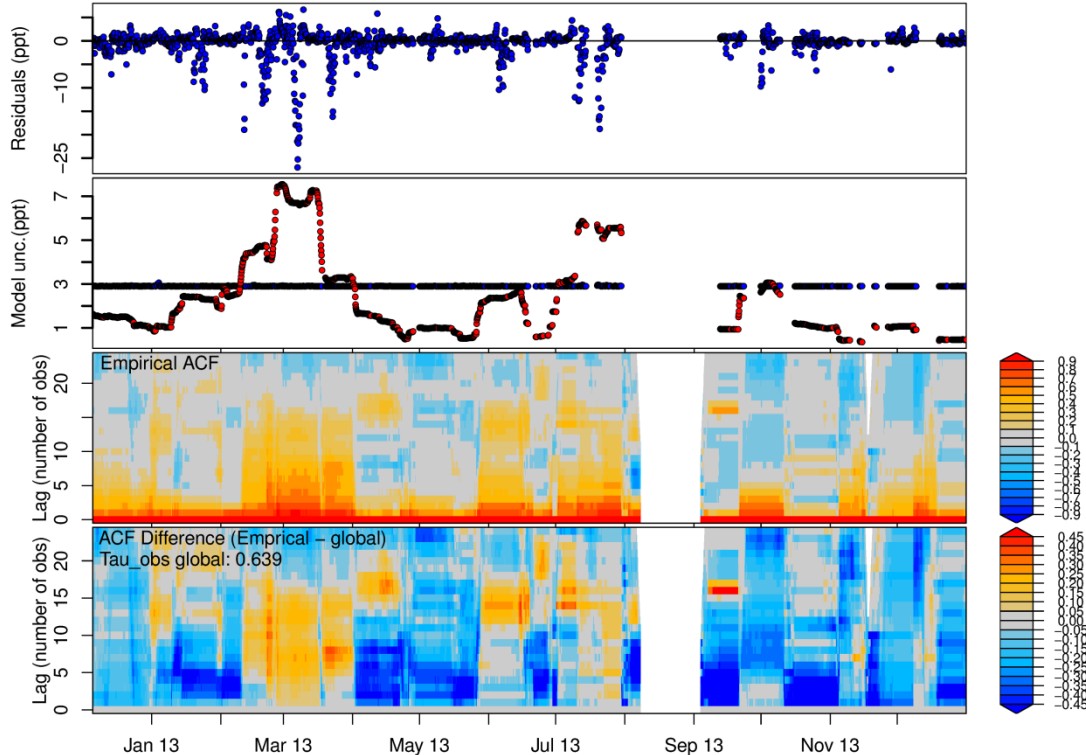

**SI-Figure 4: Time series of a) "prior" model residuals, b) data-mismatch uncertainty, blue symbols $\sigma_c$, and running**
**RMS, red symbols, c) empirical auto correlation function based on 10 day moving window, d) difference between em-**
**pirical ACF and fitted auto correlation function with constant (global) correlation length scale. All given for the site**
**MHD and for HFC-134a.**

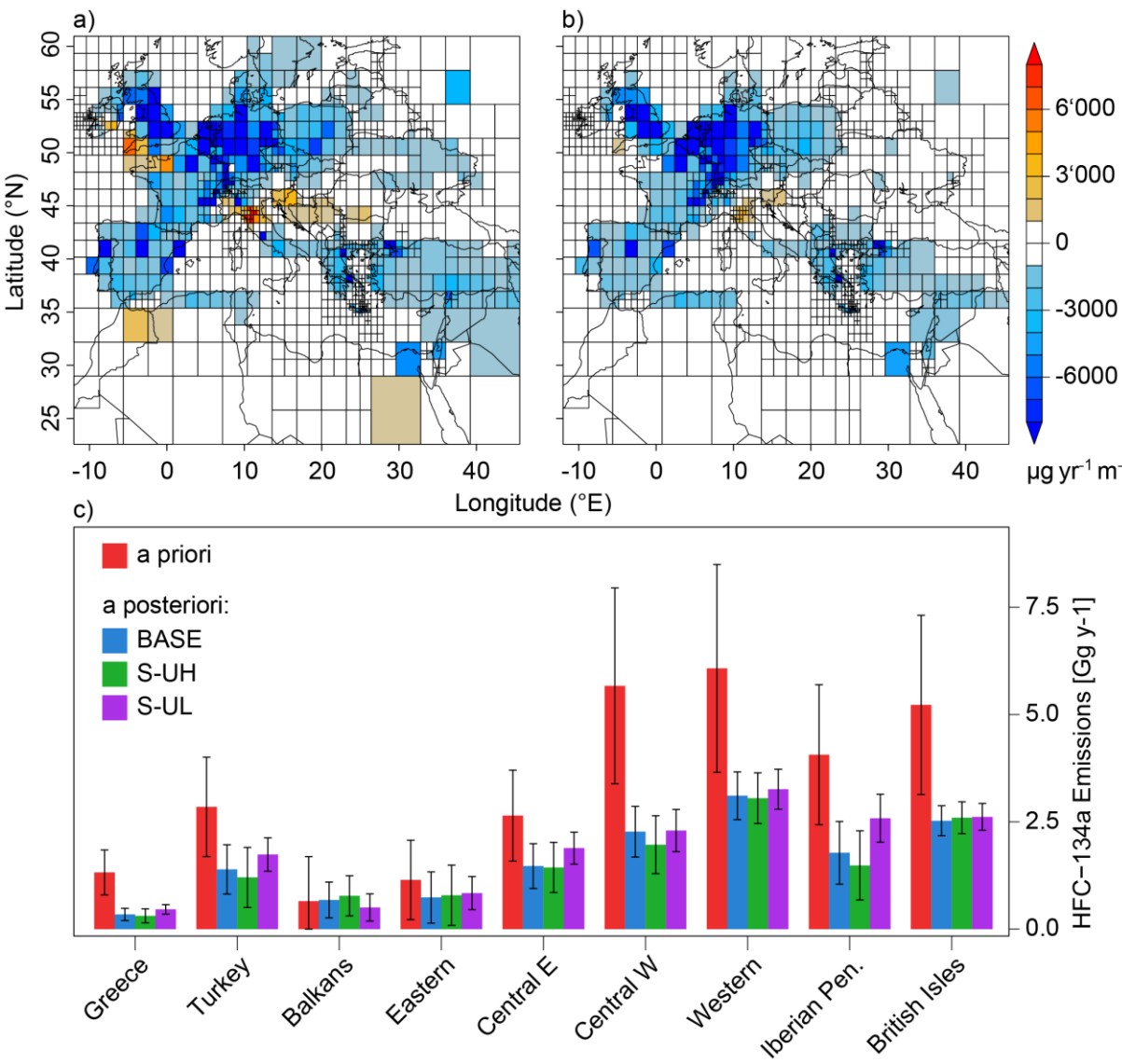

SI-Figure 5: Difference of the a posteriori and a priori emissions for (a) the S-UH and (b) the S-UL inversions. (c) regional emission estimates: a priori emissions (red) and a posteriori emissions (BASE = green, S-UH = blue, S-UL = purple). The uncertainties given are two standard deviations of the analytic uncertainty assigned to the a priori emissions and derived by the inversion as a posteriori uncertainties.

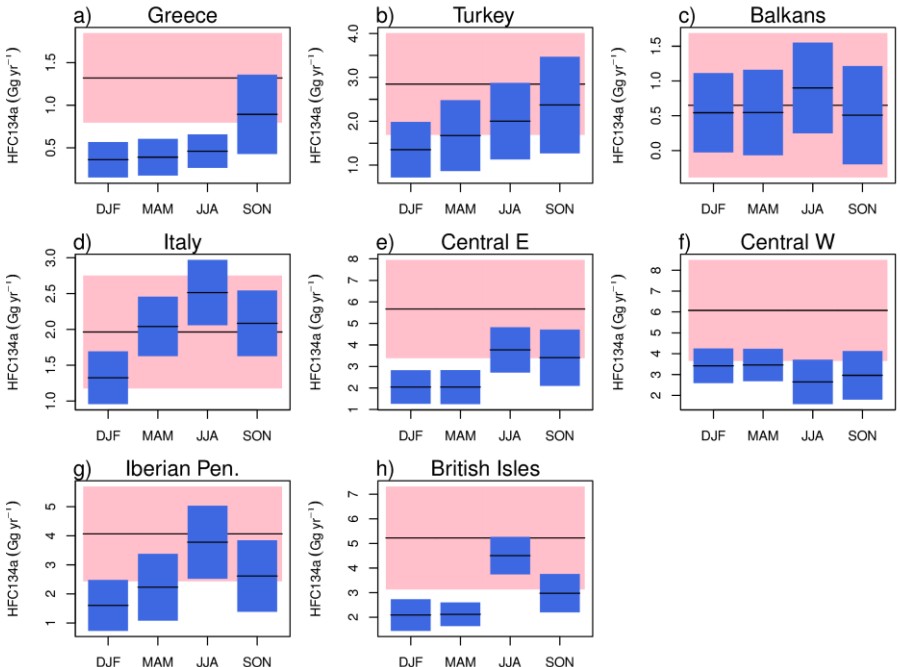

**SI-Figure 6: Seasonality of regional HFC-134a emission estimates: (red bars) a-priori and (blue bars) a posteriori emis-**
**sions. The black lines give the mean estimates and the bars denote the uncertainty (1-σ level).**