# Peer review of "Abundance and Sources of Atmospheric Halocarbons in the Eastern Mediterranean"

_Atmospheric Chemistry and Physics, 2017_

## Referee Comment (RC1) · Anonymous Referee #1 · 29 Aug 2017

General comments: The authors reported measurements of HFCs and HCFCs at a new site located in Finokalia in the Eastern Mediterranean and compared with measurements at three AGAGE sites in the western and central Europe. The authors used these measurements and estimated HFC and HCFC emissions from the Eastern Mediterranean and Europe using inverse modeling. It is nice to see that the authors considered different inversion configurations (i.e., different prior error covariances, different prior uncertainties and different observations) and tested the sensitivity of derived emissions to these different configurations. However, there are a few important tests the authors did not perform to fully demonstrate the robustness of their derived emissions: (1) the prior emissions the authors use are always biased high. I have concerns whether these biased priors may result in posterior emission estimates that

are also biased high, especially for an inversion configuration that do not consider correlated errors in the prior emissions (e.g., S-MS). The authors may consider including priors that are biased low and see if they obtain similar posterior emissions. (2) The authors derive temporally constant emissions from their inversion framework. This may result in larger model-data mismatch errors and an under-fit of their atmospheric data comparing to inversions that estimate time-varying emissions. This problem could be more severe for chemicals that have strong seasonal cycles in their emissions (e.g., HFC-134a, HCFC-22, and HFC-125). Conducting an inversion with deriving varying emissions could be beneficial to further test the robustness of their results.

Specific comments:

Lines 25-27 "The eastern Mediterranean is home to . . .under the Kyoto and Montreal Protocol". This sentence seems unnecessary in the abstract.

Lines 42-44 "reduction of the uncertainties by 40 – 80%". This reduction is for which region?

Line 51 "all long-lived halocarbons are potent greenhouse gases". This is inaccurate. Gases such as CH3Br and CH3Cl are not potent greenhouse gases.

Lines 57 – 59 "To track the development of CFC and HCFC emissions to the atmosphere,. . ." I don't think the reporting required by the MP is to specifically track emissions.

Line 71 "15% of the original value". Inaccurate. Maybe consider mentioning15% of the baseline value during the years of xx?

Line 168. Please spell out "NNE-E"

Line 183 "three-hourly aggregates". Do you mean "three-hourly averages"?

Lines 200 – 207 Transport simulation. Why do the authors choose to run their transport simulation at altitudes that are different from their actual sampling altitudes? How

much different are the derived sensitivities at their simulation altitudes and the actual sampling altitudes?

Lines 210 – 212: 1.7% degradation of HFC-152a emissions. Is this for the summer or an average for the whole year? Please be clear.

Lines 289 – 290. HCFC-142b is mostly used as a foam blowing agent whereas HCFC-22 is used as refrigerant in air conditioning. So, they are not necessarily collocated. Assuming they have the same regional emission shares is not a good assumption for creating HCFC-142b prior emissions.

Line 293. Which version of the EDGAR inventory do you mean? Be clear.

Lines 307 – 321. Uncertainty covariance matrix B. Please explain why giving correlation lengths of 200 km and 5 days? How sensitive are the derived emissions to these correlation lengths? Also be clear on whether you considered anti-correlated errors between BE and BB.

Lines 325 – 327. Calculation of $\sigma$model was derived based on the prior simulation and observations. This approach would likely overestimate the model-data mismatch errors and result in an under-fit of atmospheric data.

Lines 368 – 375. Please indicate in the paragraph whether you consider results from S-NFKL and S-OFKL into the final emission uncertainty estimation.

Lines 403 – 404. Inaccurate description on HFC-134a and HFC-125 usage. Only HFC-134a is mainly used in mobile air conditioning. HFC-125 is mainly used in commercial refrigeration and residential air conditioning.

Lines 416 – 418. It seems that the baseline was shifted at FKL for HCFC-142b in Fig. 3. What caused it?

Lines 428 – 429. "simulated a priori mole fractions reproduced the variability of the observations". Please provide the correlation (r2) value between the simulated and

observed mole fractions here. In this way, it gives quantitative information on how much variability the prior simulation explained the observed variability.

Lines 495 – 497. Error reductions were expected after an inversion. This cannot demonstrate that you achieved satisfactory emission estimates.

Lines 510 – 520. The authors discussed the differences in RMSE, R2 and Taylor Skill Scores for inversion results obtained from the "global" and "local" approaches. But how different are the covariance parameters derived from both approaches? It would be useful to show and discuss those differences first.

Lines 521 – 529. The results show the S-MS setup improves RMSE, R2 and Taylor Scores. This is, to some extent, expected and pre-defined, because the S-MS scenario did not include correlated errors in the R matrix whereas the other two scenarios (base and S-ML) did. To really test and understand the advantages or disadvantages of having correlated errors in the prior fluxes or atmospheric data, it is better to isolate the problem. In another word, it is better to have a scenario that has a same R matrix as the S-MS scenario and a same B matrix as the base scenario.

Lines 530 – 540. Results seem to indicate emissions derived from the S-MS may be more biased toward the prior.

Lines 605 – 607. It is a little dangerous to conclude a trend from two different studies given unknown differences from different methodologies and different atmospheric data.

Line 628. "account for 39.7%". Should this be ∼50% (0.53 / 1.0)?

Lines 628 – 632. "... a reduction by a factor of 2". Again, it is not a good idea to conclude a quantitative trend from two different studies.

Lines 644 – 646. Inaccurate description on the HFC usage. HFC-125 is often used with HFC-143a in commercial refrigeration or with HFC-32 as a refrigerant blend in residential air conditioning.

Lines 666 – 668. Citation of Brunner et al. [2016] here and thereafter. It is better not to cite an article that is not publicly available.

Line 695. HFC-143a is mainly used in commercial refrigeration, not in air conditioners.

Lines 717 – 719. The incomplete reporting to the UNFCCC does not seem to be an appropriate explanation for much larger reported emissions than atmosphere-derived emissions.

Lines 727 – 733. The authors compared their emission estimates with a few previous top-down emission estimates. Although the authors noted the differences, they did not provide an explanation why they are different.

Lines 806 – 808. Why including FKL increased the estimation of the baseline mixing ratios at JFJ and CMN?

Please also note the supplement to this comment: https://www.atmos-chem-phys-discuss.net/acp-2017-451/acp-2017-451-RC1-supplement.pdf

---

## Referee Comment (RC2) · Anonymous Referee #2 · 18 Sep 2017

Abundance and Sources of Atmospheric Halocarbons in the East Mediterranean

Fabian Schoenenberger et al https://doi.org/10.5194/acp-2017-451

This manuscript describes a study using high-frequency, in situ measurements of halocarbons (HCFC-22, HCFC-142b, HFC-134a, HFC-125, HFC-152a, and HFC-143a) at four sites in Europe, along with an atmospheric inversion model, to estimate regional emissions in 2013. The authors use measurements made over 6 months at Finokalia on the island of Crete (Greece), in conjunction with three other sites in Europe to obtain information on emissions sources in the Eastern Mediterranean, typically not well-captured by measurements at the other three sites (Mace Head, Ireland (MHD); Jungfraujoch, Switzerland (JFJ); Monte Cimone, Italy (CMN)). The paper describes various model sensitivity runs in which model parameters (covariance, uncertainty of

prior emissions) were varied. The authors also explored how the addition of the Finokalia measurements impacts inversion results compared to runs using data from MHD, JFJ, and CMN).

This paper makes an important contribution to previous work estimating European emissions of halogenated greenhouse gases and Montreal Protocol gases using atmospheric observations (top—down method). Measurements at Finokalia provide additional constraint and improve emissions estimates from countries in the Eastern Mediterranean, including Greece and Turkey and portions of North Africa.

General Comments:

This paper is well-written and results are well-presented.

While I don't have any problem with the methods, as the inversion technique is well-established and has been used previously to interpret similar data from MHD, JFJ, and CMN, I do have some comments regarding the results.

1) Since your period of observation is rather short (6 months), you are unable to provide any information about the seasonality of emissions. I suggest you include the potential for seasonality in your discussion. Others (Hu et al 2017; Grazioisi et al 2015, Xiang et al, 2014) have suggested that emissions of some HCFCs and HFCs show seasonal variations, with higher emissions in summer compared to winter. That, coupled with the fact that the Finokalia source sensitivity region was different between winter/spring and summer (your Figure 2), suggests that you could be missing a component of the mean annual emissions (or even the mean for 6 months) because you are less sensitive to emissions from some areas in summer, when higher emissions are more likely. This might be particularly true for Egypt. 2) You ran sensitivity studies in which the uncertainties associated with prior emissions were varied, and other sensitivity studies that explored different covariance treatments. However, you did not test the sensitivity of the magnitude of the prior emissions or distribution of priors. I think you should either include model runs with higher and lower priors, or provide some justification for why

this is not needed. Is simply changing the covariance treatment and prior uncertainty enough? 3) Can you comment on the sensitivity to background assignment? Is the choice of background more or less important compared to treatment of covariance? Did you try different background methods other than REBS? 4) Minor: It would be helpful to see (in the Supplemental) a time series similar to figure 3, but expanded over a few days or weeks. This would show more clearly the duration of "pollution" events at the different sites and provide a qualitative picture of the "signal". This would also provide some information about correlations among different halocarbon species. For example, you see what looks like pollution events for HFC-143a at Finokalia in December, but do not see corresponding pollution events for HFC-125, even though some refrigerants, such as R404A, are a blend of both compounds.

Specific Comments:

Line 50-51: CH3Cl is probably an exception to the statement that all long-lived halocarbons are potent greenhouse gases (CH3Cl lifetime is ~1 yr, but 100-yr GWP is only 11). Consider chaging "all" to "most" or "many".

Line 71: prefer "base value" to "original value"

Line 139: Is the air handling system (pump, drier) also similar to what is used at CMN? It would be helpful to specify here.

Line 202: Why not release particles at 3400 or 3500 m, closer to the actual site JFJ elevation? Is this not possible?

Line 301: Consider: "We followed three different strategies concerning the design of covariance matrices . . . . . ."

Line 429: I find it hard to see "satisfactory performance of the transport model" in figure 4. You might rephrase in terms of comparison to other studies, i.e. is this level of reproduction of variability typical for FLEXPART?

Line 467: remove comma in "driven by, an increase"
Line 495: remove comma in "which shows, that"

Line 472: Update Brunner et al 2016 to Brunner el al 2017 as this paper is now available.

Line 573: Do you mean "their mean values OR the analytical a posteriori. . .."?

Line 602: remove comma in "regions, defined by"

Line 619: maybe a comment on domain emissions compared to global total (Simmonds et al 2017) Section 3.4.2: You might consider comparing Brunner et al 2017 HFC-125 estimates for Italy (1.2 Gg with your 1.05 Gg estimate)

Line 680: comma not needed in "fact, that"

Table 3 Caption: Line 1102: " in in Mg yr-1" Duplicate word, and I think you mean "Gg yr-1"

Figure 8 caption: Is "average mean" redundant? Are uncertainties 2-sigma here also?

Figure 9 caption: Better to cite Harris and Wuebbles (2014), as GWP were calculated in Chapter 5 of the 2014 Ozone Assessment, rather than in Chapter 1, Carpenter and Reimann

[Figure]

---

## Author Comment (AC1) · 12 Dec 2017

All referee comments are given in ***bold italics***, replies in plain font.

**Anonymous Referee #1**

*General Comments:*

*The prior emissions the authors use are always biased high. I have concerns whether these biased priors may result in posterior emission estimates that are also biased high, especially for an inversion configuration that do not consider correlated errors in prior emissions (e.g., S-MS). The authors may consider including priors that are biased low and see if they obtain similar posterior emissions.*

We performed additional sensitivity inversions with 30 % lower and 30 % higher priors compared to our BASE case. Although the new low priors are still often larger than the a-posteriori values, we could not observe a strong influence of the total prior emissions onto the posterior emissions. As an indicator, the ratio between the high and the low prior sensitivity inversions was calculated for the prior and the posterior emissions. The ratio was 1.85 for the prior emissions, whereas it ranged from 0.93 to 1.25 for the posterior emissions and most regions and compounds. Consequently, in most cases the range in posterior emissions spanned by these variations in the prior was smaller than the estimated uncertainties of the posterior emissions. Exceptions were HFC-152a emissions from Greece, which were significantly larger for the high prior, and HCFC-142b emissions from Greece and Turkey. However, in the latter case the posteriori uncertainties were still larger than the range of these sensitivity runs. This clearly indicates that especially for the well simulated species the dependency on the prior emission level is not the main source of uncertainty. A discussion of these additional sensitivity inversions was included in the revised manuscript.

*The authors derive temporally constant emissions from their inversion framework. This may result in larger model-data mismatch errors and an under-fit of their atmospheric data comparing to inversions that estimate time-varying emissions. This problem could be more severe for chemicals that have strong seasonal cycles in their emissions (e.g. HFC-134a, HCFC-22, HFC-125). Conducting an inversion with deriving varying emissions could be beneficial to further test the robustness of their results.*

The seasonality of the emissions was initially not targeted, because our observations in the Eastern Mediterranean do not cover a complete annual cycle and temporally variable posterior emission estimates may suffer from this lack of observations. However, we performed one additional inversion with seasonally variable emissions of HFC-134a (the most abundant and best simulated compound). As expected, we find mixed results for the Eastern Mediterranean, where the maximum posterior emissions were derived for the fall (SON), not the summer (JJA). However, this is mainly due to the lack of observations in this period and the posterior staying close to the prior. In Western Europe we observed a clear seasonality with the expected summer peak for Italy, Germany (Central W), the Iberian Peninsula and the British Iles, but not for France and the Benelux region. Total annual emissions in the regions experiencing a seasonal cycle were slightly enhanced compared to our BASE scenario. In the revised manuscript we discuss seasonality in the emissions in general, reviewing previous studies, and specifically for HFC-134a along with the results described here.

*Specific comments:*

*Lines 25-27. "The eastern Mediterranean is home to...under the Kyoto and Montreal Protocol". This sentence seems unnecessary in the abstract.*

In our opinion, it is important to point out to the reader the diverse character of the analyzed area, due to both, countries with long established emission bans and countries where regulations have only become effective in recent years.

*Lines 42-44. "reduction of the uncertainties by 40 – 80%". This reduction is for which region?*

This number describes the range of uncertainty reduction over the entire inversion domain. For clarification purposes, the wording has been adjusted in the manuscript.

*Line 51. "all long-lived halocarbons are potent greenhouse gases". This is inaccurate.* **Gases such as CH3Br and CH3Cl are not potent greenhouse gases.**

That's correct, thank you. The wording was changed to "most" in the revised manuscript.

*Lines 57 – 59. "To track the development of CFC and HCFC emissions to the atmosphere,..." I don't think the reporting required by the MP is to specifically track emissions.*

It's correct that the reporting does only indirectly track the possible emissions of halogenated substances to the atmosphere, due to the non-emissive use of a lot of these compounds. It is not the specific reason of the reporting to the MP which is interested in the production, consumption and trade of these substances. We therefore changed this sentence in the revised manuscript to agree with the correct purpose.

*Line 71. "15% of the original value". Inaccurate. Maybe consider mentioning 15% of the baseline value during the years of xx?*

Because "original value" is very unspecific, we changed the wording to "baseline value" in the revised manuscript, according to the referee's suggestion. Due to different groups having different baseline periods and because these periods are not yet important for the results of this paper, we just mentioned the underlying 3-year baseline period though.

*Line 168. Please spell out "NNE-E"*

This was changed in the revised manuscript.

*Line 183. "three-hourly aggregates". Do you mean "three-hourly averages"?*

We used "three-hourly aggregates" due to consistency, as it has been used in other studies based on the same methods before. Because the aggregation method is based on averaging, we changed the wording to "three-hourly averages" in the revised manuscript to be more specific.

*Lines 200 – 207. Transport simulation. Why do the authors choose to run their transport simulation at altitudes that are different from their actual sampling altitudes? How much different are the derived sensitivities at their simulation altitudes and the actual sampling altitudes?*

The simulation altitudes are different from the sampling altitudes because of the terrain's representation in the underlying ECMWF model. Due to the resolution of 0.2° by 0.2° the mountainous terrain is smoothed. Therefore, the release heights should be lower than the actual sampling heights in order to catch the effect of the topography on the atmospheric flow. The release altitudes for Jungfraujoch and Monte Cimone were chosen based on tests and experience of previous studies (e.g. Keller et al., 2012), which used the same transport model and similar inversion setups. The sensitivities of Finokalia were simulated using release heights of 150 m.a.s.l. and 250 m.a.s.l. with no significant difference, which is based on the fact, that the source regions for this station are mostly located far away across the Mediterranean Sea, providing a sufficient timeframe for mixing and dispersion.

*Lines 210 – 212. 1.7% degradation of HFC-152a emissions. Is this for the summer or an average for the whole year? Please be clear.*

As mentioned in the manuscript this calculation is based on an average lifetime of HFC-152a and is therefore to be understood as an average degradation rate during the 10-day transport period, not specifically during summer. We clarified this in the revised manuscript.

*Lines 289 – 290. HCFC-142b is mostly used as a foam blowing agent whereas HCFC-22 is used as refrigerant in air conditioning. So, they are not necessarily collocated. Assuming they have the same regional emission shares is not a good assumption for creating HCFC-142b prior emissions.*

In the absence of better alternatives, we used the collocation of HCFC-142b and HCFC-22 based on the theoretical assumption, that emissions of these compounds are likely to be collocated on a larger scale (order of 100 km) due to existing regulations.

*Line 293. Which version of the EDGAR inventory do you mean? Be clear.*

We used EDGAR v4.2 and EDGAR v4.2 FT2010. The description was changed in the respective paragraphs and in the references.

*Lines 307 – 321. Uncertainty covariance matrix B. Please explain why giving correlation lengths of 200 km and 5 days? How sensitive are the derived emissions to these correlation lengths? Also be clear on whether you considered anti-correlated errors between BE and BB.*

The spatial correlation length scale was chosen based on findings in previous studies where a maximum likelihood optimization was used to estimate covariance parameters (Brunner el al., 2012, Henne et al., 2016). In this study, no such parameter optimization could be carried out, therefore, we followed the range used in previous work. The spatial correlation length scale does not largely impact region total emissions when varied within reasonable range (100 – 500 km).  Lower values often lead to undesired dipole patterns in the posterior adjustments, whereas larger values lead to very smooth patterns that do not allow the inversion to pick up differences in neighboring regions. 200 km seems to present a reasonable value in the golden middle. Similar arguments can be found for the temporal correlation length of the baseline. One important argument for this value is the fact that synoptic systems, which are responsible for the advection of different baseline concentrations, exhibit time scale of about 5 days as well. In the case of the baseline concentrations of the compounds discussed here the influence of the correlation time scale is not very large because the baseline uncertainty itself is relatively small, so that adjustments to the baseline are also relative small and do not alter strongly with the temporal correlation time scale. No covariance between emissions and baseline were considered in the prior. However, we checked the posterior covariance between emissions and baseline to rule out that large negative covariance existed.

*Lines 325 – 327. Calculation of σ model was derived based on the prior simulation and observations. This approach would likely overestimate the model-data mismatch errors and result in an under-fit of atmospheric data.*

The procedure to estimate $\sigma_{model}$ was not described correctly in the manuscript. Actually, we are not using the RMSE from the prior simulation, but iterate the inversion 3 times using the RMSE from the previous posterior simulation in the current iteration while keeping all other parameters and prior fields the same. This approach is similar to that suggested by Stohl et al. (2009) and results in a more reasonable model-data mismatch error. The description was updated in the revised manuscript.

*Lines 368 – 375. Please indicate in the paragraph whether you consider results from S-NFKL and S-OFKL into the final emission uncertainty estimation*

The results of S-NFKL and S-OFKL were NOT used in the final uncertainty emissions but were solely run to evaluate the benefit of additional measurements in Finokalia and the influence of additional measurements at the existing AGAGE sites. This was already pointed out in the original manuscript in section 3.4 (L572-580) but is now mentioned at this location in the revised manuscript.

*Lines 403 – 404. Inaccurate description on HFC-134a and HFC-125 usage. Only HFC-134a is mainly used in mobile air conditioning. HFC-125 is mainly used in commercial refrigeration and residential air conditioning.*

This was corrected in the respective paragraph of the revised manuscript.

*Lines 416 – 418. It seems that the baseline was shifted at FKL for HCFC-142b in Fig.3. What caused it?*

We assume that the change of the general direction of advection caused the shift in the baseline of HCFC-142b. However this doesn't have a large impact on the determination of regional emission values, as the baseline for each site is adjusted during the inversion and only the signal above the baseline is relevant.

*Lines 428 – 429. "simulated a priori mole fractions reproduced the variability of the observations". Please provide the correlation (r2) value between the simulated and observed mole fractions here. In this way, it gives quantitative information on how much variability the prior simulation explained the observed variability*

The coefficient of determination ($R^2$) and also the difference between simulated signals of the prior and the posterior simulation for the four sites is discussed in the following paragraph, showing that the inversion performance is comparable to other studies with similar inversion schemes (e.g. Stohl, et al., 2009; Keller, et al., 2012). In this paragraph, a reference to Table 2 is given, which contains all the $R^2$ values. To make this clearer, a reference to Table 2 has been added to the first paragraph mentioned by the referee.

*Lines 495 – 497. Error reductions were expected after an inversion. This cannot demonstrate that you achieved satisfactory emission estimates.*

We are aware that uncertainty reductions are not a proof of satisfactory posterior emission estimates. However, these error reductions are an indication that the inversion is working correctly which we tried to express with this sentence. Because we understand, that this sentence does not improve the understanding but is rather confusing, we deleted it in the revised manuscript.

*Lines 510 – 520. The authors discussed the differences in RMSE, R2 and Taylor Skill Scores for inversion results obtained from the "global" and "local" approaches. But how different are the covariance parameters derived from both approaches? It would be useful to show and discuss those differences first.*

We agree with the reviewer that the differences in the covariance matrices were not sufficiently discussed. The main difference, as mentioned in the methods section, was the scope of the temporal correlation length scale, $\tau_c$. In the "global" method only a single correlation length scale was used for each site, whereas for the "local" approach a local correlation length scale was used the varied for each time and site and was based on a local window of 10 days. The variances on the diagonal of the data-mismatch covariance matrix were the same in both methods; at least for the first iteration of the RMS estimator. Since the iteration uses the posterior emissions for a new estimate of the RMS error, the final values may be different for the local and global method. We added this detail, which we had missed in the original version, in the revised manuscript (see also comment above on RMS

error). Having said this, one can first compare the total covariance by site that is contained in **R** by calculating

$$\sigma_k = \sqrt{\frac{R_k \cdot R_k^T}{N_k}},$$

where **R$_k$** is the block matrix belonging to all N$_k$ observations/simulations of a single site.

In the case of the HFC-134a inversion, discussed at the mentioned location in the text, $\sigma_k$ took values of 4.2, 7.3, 8.7 and 8.7 ppt for our BASE inversion (global $\tau_c$) and the sites FKL, JFJ, MHD and CMN, respectively. For the S-ML sensitivity inversion (local $\tau_c$) these values only differed slightly for the sites FKL and CMN, but were 8.3 and 9.0 for the sites JFJ and MHD, respectively. As a consequence less (more) weight was given to the observations from JFJ (MHD) in S-ML than in the BASE inversion. Especially for MHD one would expect that the posterior performance would be increased in the S-ML case compared to the BASE inversion. As described in the original text, this was not the case. The reason can be found in the distinctly different temporal pattern of the temporal correlation length scale. Panel d) of the figure below shows the differences between the empirical auto correlation function for a running window width of 10 days and the fitted auto correlation function with a constant (global) correlation length scale for the site MHD. MHD infrequently receives pollution events from the European continent. These episodes are usually characterized by relatively large model residuals (panel a). Also the auto correlation of the residuals during these periods is enhanced (panel c). The global estimate of $\tau_c$ then leads to an underestimation of auto-correlation during these periods (indicated by positive values in panel d). Finally, this means that in the BASE inversion more weight (smaller auto correlation, and, hence, smaller covariance) is given to the observations from MHD during the pollution events as compared to the sensitivity inversion with local $\tau_c$. In turn, the posterior adjustments for MHD have a larger impact for the BASE inversion and performance improves more than in the S-ML case.

The above discussion for our example compound HFC-134a was added to the revised manuscript and the plot added to the supplementary material.

[Figure]

**Figure 1: Time series of a) "prior" model residuals, b) data-mismatch uncertainty, blue symbols $\sigma_c$, and running RMS, red symbols, c) empirical auto correlation function based on 10 day moving window, d) difference between empirical ACF and fitted auto correlation function with constant (global) correlation length scale. All given for the site MHD and for HFC-134a.**

*Lines 521 – 529. The results show the S-MS setup improves RMSE, R2 and Taylor Scores. This is, to some extent, expected and pre-defined, because the S-MS scenario did not include correlated errors in the R matrix whereas the other two scenarios (base and S-ML) did. To really test and understand the advantages or disadvantages of having correlated errors in the prior fluxes or atmospheric data, it is better to isolate the problem. In another word, it is better to have a scenario that has a same R matrix as the S-MS scenario and a same B matrix as the base scenario.*

We thank the reviewer for this comment and addressed his concern in the discussion. However, we could not change our previous setup in the limited time available for the revision of this paper but will address this issue in future analysis.

*Lines 530 – 540. Results seem to indicate emissions derived from the S-MS may be more biased toward the prior.*

This seems to be unlikely, since the total data-mismatch covariance for the S-MS was the lowest of all three sensitivity inversions and, hence, the posterior simulations should be closest to the observations in this case, which agrees with the performance analysis in the manuscript. The reason for relatively large posterior emissions (closer to the prior) more likely is due some unrealistic adjustments of individual grid cells that are not well constrained due to the absence of off-diagonal elements in the covariance matrices.

*Lines 605 – 607. It is a little dangerous to conclude a trend from two different studies given unknown differences from different methodologies and different atmospheric data.*

We completely agree with the referee, that a trend cannot be deduced from 2 data points. To avoid making readers think that we indicate a trend based on our findings, we changed the sentence using

a less "impactful" wording to still show the differences in these two studies and what this "could" possibly mean.

**Line 628. "account for 39.7%". Should this be ~50% (0.53 / 1.0)?**

Right, the wrong numbers have mistakenly been used to calculate those percentages. We updated the values in the revised manuscript.

**Lines 628 – 632. "...a reduction by a factor of 2". Again, it is not a good idea to conclude a quantitative trend from two different studies.**

It was not our intention to show a quantitative trend in actual emissions here but to show the difference of the estimated values between our study and the one by Keller et al. (2012). To further avoid readers to think about a proposed trend, the wording was changed in the revised manuscript.

**Lines 644 – 646. Inaccurate description on the HFC usage. HFC-125 is often used with HFC-143a in commercial refrigeration or with HFC-32 as a refrigerant blend in residential air conditioning.**

The description was adjusted in the revised manuscript.

**Lines 666 – 668. Citation of Brunner et al. [2016] here and thereafter. It is better not to cite an article that is not publicly available.**

Meanwhile the article has been published and changes made to the discussion paper do not affect our manuscript. We now cite the finally published article and changed citations and the bibliography in the revised manuscript accordingly.

Brunner, D., T. Arnold, S. Henne, A. J. Manning, R. L. Thompson, M. Maione, S. O'Doherty, and S. Reimann  (2016). Comparison of four inverse modelling systems applied to the estimation of HFC-125, HFC-134a and SF6 emissions over Europe, Atmos. Chem. Phys., 17(17), 10651-10649, doi: 10.5194/acp-17-10651-2017.

**Line 695. HFC-143a is mainly used in commercial refrigeration, not in air conditioners.**

The description was changed in the revised manuscript.

**Lines 717 – 719. The incomplete reporting to the UNFCCC does not seem to be an appropriate explanation for much larger reported emissions than atmosphere-derived emissions.**

The sentence has been reformulated to avoid the impression, that incomplete reporting to the UNFCCC is the explanation for much larger reported emissions than the ones derived from atmospheric measurements. The main focus is on the issue, that emissions are reported in the country of manufacturing, although most of the emissions may occur during the usage of the product, which does not necessarily have to be the same country.

**Lines 727 – 733. The authors compared their emission estimates with a few previous top-down emission estimates. Although the authors noted the differences, they did not provide an explanation why they are different.**

Implicitly it is assumed, that any differences in top-down emission estimates originate in emission changes. However, we are aware that with the current network of three routine monitoring sites in Europe, complemented with the observations at FKL, it is difficult to determine smaller scale emission patterns. It was also shown by Brunner et al. (2017) that the use of different inversion systems can lead to significantly different results on the country scale, which makes the comparison of different European studies difficult, as long as there is no denser measurement network available.

***Lines 806 – 808. Why including FKL increased the estimation of the baseline mixing ratios at JFJ and CMN?***

FKL is sensitive to emissions from Italy, given the appropriate flow conditions. As a consequence, including measurements in FKL will impact the estimated regional emissions from Italy. A change in emissions from Italy, which do influence the simulations at JFJ and especially CMN, can thus have an influence on the baseline of these stations as well. Hence, the occurrence of a long-distance effect of the measurements in FKL on the whole inversion system can be understood qualitatively. However, we cannot assess if the magnitude of the effect is correct.

---

## Author Comment (AC2) · 12 Dec 2017

All referee comments are given in **_bold italics_**, replies in plain font.

**Anonymous Referee #2**

*General Comments:*

*Since your period of observation is rather short (6 months), you are unable to provide any information about the seasonality of emissions. I suggest you include the potential for seasonality in your discussion. Others (Hu et al 2017; Grazioisi et al 2015, Xiang et al, 2014) have suggested that emissions of some HCFCs and HFCs show seasonal variations, with higher emissions in summer compared to winter. That, coupled with the fact that the Finokalia source sensitivity region was different between winter/spring and summer (your Figure 2), suggests that you could be missing a component of the mean annual emissions (or even the mean for 6 months) because you are less sensitive to emissions from some areas in summer, when higher emissions are more likely. This might be particularly true for Egypt.*

We added an analysis and discussion of seasonality in the revised manuscript following the suggestions of both referees. The limited observation period in in the Eastern Mediterranean and the seasonality of the transport patterns in the same area make it difficult to derive certain results on seasonality here.

*You ran sensitivity studies in which the uncertainties associated with prior emissions were varied, and other sensitivity studies that explored different covariance treatments. However, you did not test the sensitivity of the magnitude of the prior emissions or distribution of priors. I think you should either include model runs with higher and lower priors, or provide some justification for why this is not needed. Is simply changing the covariance treatment and prior uncertainty enough?*

As also suggested by referee #1 we added two sensitivity runs that explore the sensitivity to the absolute prior emissions. The results indicate only a minor dependency on the prior emissions with a few exceptions. See also reply to referee #1. We included a discussion of these two new sensitivity runs in the revised manuscript.

*Can you comment on the sensitivity to background assignment? Is the choice of background more or less important compared to treatment of covariance? Did you try different background methods other than REBS?*

The temporal baseline variability for HFCs and HCFCs is generally small and the pollution peaks above baseline to estimate emissions are comparably high. Of course, the baseline is important to calculate regional emission estimates (see for example Brunner et al., 2017). However because we chose the setup such that the baseline is readjusted during the inversion, the algorithm is not similarly sensitive to the initial background assignment and the treatment of covariance is more important in this case. Because of this, we did not go into further detail concerning the background method and only used REBS, which was successfully used for similar studies already in the past.

*Minor: It would be helpful to see (in the Supplemental) a time series similar to figure 3, but expanded over a few days or weeks. This would show more clearly the duration of "pollution" events at the different sites and provide a qualitative picture of the "signal". This would also provide some information about correlations among different halocarbon species. For example, you see what looks like pollution events for HFC-143a at Finokalia in December, but do not see corresponding pollution events for HFC-125, even though some refrigerants, such as R404A, are a blend of both compounds.*

Our experience from other sites is that even though that blends are important sources, pollution events can not be compared generally. In the case mentioned here for December no pollution event can be seen for HFC-125, because the measurements could not be analysed correctly due to an overlapping unknown peak and, hence, were flagged invalid and excluded from further analysis.

Nevertheless, to underline the duration of pollution events and the correlation between different compounds, a plot like figure 3 concentrating just on the observations in June (best data coverage) was added to the supplementary material and is discussed in the revised manuscript.

***Specific comments:***

***Line 50-51: CH3Cl is probably an exception to the statement that all long-lived halocarbons are potent greenhouse gases (CH3Cl lifetime is ~ 1 yr, but 100-yr GWP is only 11). Consider changing "all" to "most" or "many".***

We changed this to "most" in the revised manuscript.

***Line 71: prefer "base value" to "original value"***

Changed to "baseline value" and described how this baseline values are determined as requested by referee #1 too.

***Line 139: Is the air handling system (pump, drier) also similar to what is used at CMN? It would be helpful to specify here.***

Although some minor parts may be slightly different, the instruments at CMN and the one we used are practically identical. We now specified this in the paragraph as well.

***Line 202: Why not release particles at 3400 or 3500 m, closer to the actual site JFJ elevation? Is this not possible?***

It is possible to release particles on any chosen height. The determination of a lower release height than the actual height of the JFJ site in the model environment is based on the smoothed terrain because of the limited resolution of the underlying model terrain. To avoid releasing particles too high above model ground, a lower release altitude is used. For a more detailed description we refer to the same question asked by referee #1.

***Line 301: Consider: "We followed three different strategies concerning the design of covariance matrices ......***

Good suggestion which we like to use, thank you.

***Line 429: I find it hard to see "satisfactory performance of the transport model" in figure 4. You might rephrase in terms of comparison to other studies, i.e. is this level of reproduction of variability typical for FLEXPART?***

The performance visible in Figure 4 and statistically summarized in Table 2 is very typical for this kind of application of the FLEXPART transport model to regional-scale halocarbon simulations. We added a set of references to publications that had achieved similar performance in the past.

***Line 467: remove comma in "driven by, an increase"***

Removed in the revised manuscript.

***Line 495: remove comma in "which shows, that"***

Removed in the revised manuscript.

*Line 472: Update Brunner et al 2016 to Brunner el al 2017 as this paper is now available.*

Because the article has meanwhile been published and changes made to the discussion paper do not affect our manuscript, we now cite the published article and changed citations and the bibliography of the revised manuscript accordingly.

*Line 573: Do you mean "their mean values OR the analytical a posteriori ...."?*

Referring to Figure 7, the first measure we use is the range of the mean values of the a posteriori emission estimates (box) of the incorporated BASE and sensitivity runs. The average of these mean values is depicted with the thick horizontal line. The error bars give the analytical uncertainty (95% confidence level) averaged over all uncertainty inversions.

In the text and table, the values are given as the average of all the mean values of the BASE and sensitivity runs. The uncertainty is the respective maximum/minimum value derived from all the mean values of these 5 inversions AND the analytical uncertainty (95 % confidence interval).

*Line 602: remove comma in "regions, defined by"*

Removed in the revised manuscript.

*Line 619: maybe a comment on domain emissions compared to global total (Simmonds et al 2017) Section 3.4.2: You might consider comparing Brunner et al 2017 HFC-125 estimates for Italy (1.2 Gg with your 1.05 Gg estimate)*

A comment to both global and Italian emissions in Simmonds et al. 2017 and Brunner et al. 2017 were added to the revised manuscript.

*Line 680: comma not needed in "fact, that"*

Removed in the revised manuscript.

*Table 3 Caption: Line 1102: " in in Mg yr-1" Duplicate word, and I think you mean "Gg yr-1"*

Changed in the revised manuscript.

*Figure 8 caption: Is "average mean" redundant? Are uncertainties 2-sigma here also?*

It's the average of all 5 mean values, given by each individual inversion, similar to the definition used in Figure 7, therefore not redundant. Again we use the "structural uncertainty (given by the range of mean values of each inversion run) and the "analytical uncertainty" (given as 95% confidence interval/2-sigma uncertainty) to define a total error here.

*Figure 9 caption: Better to cite Harris and Wuebbles (2014), as GWP were calculated in Chapter 5 of the 2014 Ozone Assessment, rather than in Chapter 1, Carpenter and Reimann*

We are now citing Harris and Wuebbles (2014) instead of Carpenter and Reimann in caption of Figure 8.

---

## Author Response (AR2)

**Abundance and Sources of Atmospheric Halocarbons in the Eastern Mediterranean**

Fabian Schoenenberger[1], Stephan Henne[1], Matthias Hill[1], Martin K. Vollmer[1], Giorgos Kouvarakis[2], Nikolaos Mihalopoulos[2], Simon O'Doherty[3], Michela Maione[4], Lukas Emmenegger[1], Thomas Peter[5], Stefan Reimann[1]

[1]Laboratory for Air Pollution/Environmental Technologies, Empa, Swiss Federal Laboratories for Materials Science and Technology, Dübendorf, Switzerland
[2]Department of Chemistry, University of Crete, Heraklion Crete, Greece
[3]School of Chemistry, University of Bristol, Bristol, UK
[4]Department of Pure and Applied Sciences, University of Urbino, Urbino, Italy
[5]Institute for Atmospheric and Climate Science, ETH Zürich, Zürich, Switzerland

*Correspondence to*: Stephan Henne (stephan.henne@empa.ch)

**Author's Response**

Dear Editor,

Thanks to the valuable comments and suggestions, we were able to further clarify our work.

We slightly extended the discussion concerning the emission ratios of HFC-152a for the high and low a priori sensitivity inversions. All other more minor comments were followed in the revised manuscript and are detailed in the following point-by-point response to the Co-Editor.

We believe that with the additional corrections the manuscript is now fit for final publication in ACP and are looking forward to your response.

Sincerely,
Fabian Schönenberger

All Co-Editor comments are given in ***bold italics***, replies in plain font.

***p.3/l.71: The regulations of HFCs in the Kigali amendment are actually rather complicated. I suggest to either explain them in detail or (my suggestion) just state that HFCs are now included in the MP, but that no immediate reductions are foreseen.***

We adapt the Co-Editors suggestion and just state that in 2016, HFCs were included in the Montreal Protocol by the Kigali Amendment. However, we add the step-wise phase down of consumption as an additional general statement. We would not mention that no immediate actions are foreseen, as in fact the baseline for Non-A5 countries has already passed (2011-2013) and first reductions will take place in 2019, which in our opinion is quite "immediate".

***p.21/l.635: Should this be emissions instead of sensitivities?***

We use the term "sensitivity inversion" to describe the inversions performed to analyse the robustness of our model towards different variables. Thus, in this context "sensitivity" is correct. However, as this sentence is slightly confusing we revised it as follows:

"As an indicator, the ratios between the a priori and a posteriori emissions were calculated for the sensitivity inversions using high and low a priori emissions."

***p.21/l.636: You might want to point out that the ratio of 1.85 is actually prescribed***

We now point out that the ratio for the a priori emissions is prescribed.

***p.21/l.640: Could you comment more on the differences derived for HFC-152a? Are they outside the estimated uncertainties?***

We slightly extended the discussion of the emission ratios for HFC-152a for the high and low a priori sensitivity inversions. It now reads as follows:

Exceptions to this reduction in the ratio between high and low a posteriori emissions were HFC-152a emissions from Greece (a posteriori ratio of 2.4). In this case a posteriori emissions were significantly larger for the high a priori inversion than for the base and low a priori inversion. Furthermore, the ratio only slightly decreased for HCFC-142b emissions from Greece and Turkey (a posteriori ratio of 1.6).

As for the reasons of the increase in the ratio for HFC-152a we could only give very unsure speculations and, therefore, did not further comment on it.

***p.27/l.830: This sentence is a bit ambiguous. I suggest rephrasing: …emissions of HFC-152a are reported in the country where the consumer product is manufactured, not in the country where emissions are likely to occur during use or disposal.***

The Co-Editors suggestion to clarify this sentence was adapted. However, we prefer a slightly stronger emphasis on the fact that these emissions occur in the real world. The wording was therefore changed as follows:

"…, emissions of HFC-152a are reported in the country where the consumer product is manufactured, not in the country where emissions are occurring during use or disposal."

***p.31/l.951: I suggest to replace "are largely independent" by a statement that the estimates sty with the quoted uncertainty range.***

We don't like to replace this statement because it summarises the important result that our inversion, though based on a priori knowledge, was not too sensitive to the choice of a priori parameters. Or in other words there was sufficient observational constraint to derive useful a posteriori results. However, we added the following statement, that for most compounds and inversion regions the derived analytical a posteriori uncertainty was similar to the spread derived from the sensitivity inversions (see Figure 7).

"Hence, for most compounds and emission regions the derived analytical a posteriori uncertainty was similar to the spread of the a posteriori emissions from all sensitivity inversions."

***Additional comment: Please include somewhere (e.g. Figure caption where it is first used) that ppt refers to pmol mol$^{-1}$, which is the SI unit.***

As suggested by the Co-Editor, a referral of ppt to its SI unit was included in Figure 3, where it is first used.

[revised manuscript text omitted]